# Contextual Subspace Approximation with Neural Householder Transforms

## Abstract

Choosing an appropriate action representation is an integral part of solving robotic manipulation problems. Published approaches include latent action models which compress the control space into a low dimensional manifold. These involve training a conditional autoencoder, where the current observation and a low-dimensional action are passed through a neural network decoder to compute high dimensional actuation commands. Such models can have a large number of parameters, and can be difficult to interpret from a user perspective. In this work, we propose that similar performance gains in robotics tasks can be achieved by restructuring the neural network to map observations to a basis for a context-dependent linear actuation subspace. This results in an action interface wherein a user's actions determine a linear combination of a state-conditioned actuation basis. We introduce the Neural Householder Transform (NHT) as a method for computing this basis. Our results suggest that reinforcement learning agents trained with NHT in kinematic manipulation and locomotion environments are more robust to hyperparameter choice and achieve higher final success rates compared to agents trained with alternative action representations. NHT agents outperformed agents trained with joint velocity/torque actions, agents trained with an SVD actuation basis, and agents trained with a LASER action interface in the WAMWipe, WAMGrasp, and HalfCheetah environments.

## 1 Introduction

In real-world applications of reinforcement learning, its imperative to choose appropriate representations when defining the Markov decision process. The consequences of poor design decisions can have adverse effects in domains like robotics, where safety (Tosatto et al., 2021) and sample efficiency (Li et al., 2021) are desirable properties. Typically these properties can be captured by choice of action space. Choices of robot action types distinct from basic joint motor control, such as Cartesian control or impedance control, have been shown to influence the efficiency of robotic learning, depending on the task (Martín-Martín et al., 2019).

Researchers have typically focused on learning action representations that can capture a variety of robotic motions. This interest has led to developing several different action representation frameworks. One framework includes motor primitives in which entire trajectories are encoded as the action (Paraschos et al., 2013; Schaal, 2006). Motor primitives have seen much success in robotics leading to impressive real-world experimental results by constraining the action space (Tosatto et al., 2021; Kober & Peters, 2009).

Another framework is the *latent actions* framework, in which actions-per-time-step are compressed into a latent subspace. Typically these are conditional auto-encoders trained to predict the high-dimensional actions given the state and latent action. These methods have been used successfully in both learning systems (Allshire et al., 2021; Zhou et al., 2020; van der Pol et al., 2020) as well as human-in-the-loop settings (Losey et al., 2021; 2020; Karamcheti et al., 2021; Jun Jeon et al., 2020).

It remains unclear whether robotics tasks must have deep, complex action models. There is little work comparing latent action models across varying complexity tasks. For example, hand poses - a complex high dimensional action space - can have up to 80% of configurations explained by two principal components (Santello et al., 1998). This result has been exploited to develop low-dimensional linear control algorithms, but they assume all actions exist in a global linear subspace

(Matrone et al., 2012; Odest & Jenkins, 2007; Artemiadis & Kyriakopoulos, 2010; Liang et al., 2022).

In this work we propose an approach in which we use a neural network to produce a state-dependent basis for a linear actuation subspace. We refer to this as *contextual subspace approximation*. Actuation commands (e.g. joint velocities) are locally linear with respect to low dimensional inputs, but globally non-linear as the actuation subspace changes as a function of context.

The motivation for contextual subspace approximation and the corresponding solutions can be summarized as follows: 1) Contextual subspace approximation requires less data because a $k$-dimensional subspace is completely determined by just $k$ linearly independent samples. 2) From the agent's perspective, action maps change the transition dynamics of the environment, and using simpler functions results in simpler dynamics. 3) Models for contextual subspace approximation can be notably smaller by doing away with the encoder from the latent actions framework.

The model proposed here uses Householder transformations to obtain an orthonormal basis for the desired actuation subspace. Householder transformations are often used in QR factorization to efficiently compute least square solutions to over-determined systems of linear equations. This property has been exploited in several settings to define learnable orthonormal matrices in applications of QR factorization for machine learning (Nirwan & Bertschinger, 2019; Dass & Mahapatra, 2021; van den Berg et al., 2018). Additional work has studied applications of Householder reflections that include normalizing flows (Tomczak & Welling, 2016; Mathiasen et al., 2020), network activation functions (Singla et al., 2021), and decomposition of recurrent and dense layers in neural networks (Mhammedi et al., 2017; Zhang et al., 2018; Obukhov et al., 2021). To the best of our knowledge, our work is the first to study Householder matrices in the context of latent action models.

We identify our contributions as the following:

- We propose contextual subspace approximation as a novel alternative to end-to-end non-linear latent action models for robotic control.
- We prove that the Neural Householder Transform is smooth with respect to changes in context, and can output bases for the optimal actuation subspace associated with each context.
- Our experiments empirically suggest that in two simulated kinematic manipulation tasks and one locomotion task, reinforcement learning agents trained with Neural Householder Transforms learn more efficiently than agents trained to act in with 7dof, SVD, or LASER action interfaces.

## 2 BACKGROUND AND PRELIMINARIES

In this section, we formalize our framework for learning action representations. We outline relevant background knowledge to contextualize our work, including deep latent action models, and their combination with Markov decision processes. We compare linear, locally-linear, and nonlinear action mapping approaches by conducting experiments on reinforcement learning problems.

### 2.1 PROBLEM STATEMENT

We assume that the data we wish to model was observed in some *context*, and the resulting dataset is a collection of context-datapoint pairs (datapoints and context are both represented by vectors). We formulate the problem of **contextual subspace approximation** by supposing that, for every context $\mathbf{c}$, there exists an associated subspace that best approximates the data observed in the neighborhood of $\mathbf{c}$.

We use $\mathbf{x} = (\mathbf{c}, \mathbf{u})$ to denote a tuple consisting of a datapoint $\mathbf{u}$ and the context $\mathbf{c}$ in which it was observed. For convenience, we define the following functions to extract the data and context from a tuple $\mathbf{x}$, respectively: $\mathcal{C}(\mathbf{x}) = \mathbf{c}$; $\mathcal{U}(\mathbf{x}) = \mathbf{u}$. In addition, we denote the neighborhood of a context point as $\mathcal{N}(\mathbf{c}) = \{\mathbf{c}' : \|\mathbf{c} - \mathbf{c}'\| < \delta\}$ for some $\delta \in \mathbb{R}$.

**Definition 2.1** (Optimal Contextual Subspace). We define $W^*(\mathbf{c})$, the optimal $k$-dimensional subspace associated with context $\mathbf{c}$, as the $k$-dimensional subspace that minimizes the expected projection error of data observed in the neighborhood of $\mathbf{c}$:

$$W^*(\mathbf{c}) \doteq \arg\min_W \mathbb{E}_{\mathbf{x}|\mathcal{C}(\mathbf{x})\in\mathcal{N}(\mathbf{c})} \|\mathcal{U}(\mathbf{x}) - \mathrm{proj}_W\left(\mathcal{U}(\mathbf{x})\right)\|_2^2 \qquad (1)$$

where $W$ is a $k$-dimensional linear subspace of $\mathbb{R}^n$, and $\mathrm{proj}_W(\mathcal{U}(\mathbf{x}))$ is the orthogonal projection of the data $\mathbf{u}$ onto $W$.

Our goal is to approximate a function $\mathcal{Q}^*(\mathbf{c})$ that maps context vectors to an orthonormal basis for the associated optimal contextual subspace.

$$\mathcal{Q}^* : \mathbf{c} \mapsto \hat{\mathbf{Q}} \mid \mathrm{col}(\hat{\mathbf{Q}}) = W^*(\mathbf{c}) \qquad (2)$$

where $\hat{\mathbf{Q}} \in \mathbb{R}^{n \times k}$ is a $n \times k$ matrix of real numbers, and $\mathrm{col}(\hat{\mathbf{Q}})$ is the column space of $\hat{\mathbf{Q}}$. We assume access to a dataset of datapoint-context pairs.

## 2.2 MARKOV DECISION PROCESSES

A Markov Decision Process (MDP) is defined by the tuple $\langle \mathbb{S}, \mathbb{A}, T, p(\mathbf{s}_0), r(\mathbf{s}, \mathbf{a}, \mathbf{s}') \rangle$ where $\mathbb{S}$ is the state space and $\mathbb{A}$ is the action space. The transition probability operator $T(\mathbf{s}, \mathbf{a}, \mathbf{s}') : \mathbb{S} \times \mathbb{A} \times \mathbb{S} \to [0, 1]$ denotes the probability of transitioning to state $\mathbf{s}' \in \mathbb{S}$ when taking an action $\mathbf{a} \in \mathbb{A}$ from a state $\mathbf{s} \in \mathbb{S}$. $p(\mathbf{s}_0)$ is the initial state distribution, and $r(\mathbf{s}, \mathbf{a}, \mathbf{s}')$ defines the reward function. In this framework, the optimal policy search problem involves finding some $\pi^*(\mathbf{s})$ that maximizes the discounted return: $\pi^*(\mathbf{s}) = \arg\max_\pi V_\pi(\mathbf{s}) = \mathbb{E}[\sum_{i=t}^T \gamma^t r(\mathbf{s}, \mathbf{a}, \mathbf{s}')]$.

Often in real-world problems, reinforcement learning agents must approximate $\pi^*(\mathbf{s})$ as the state representation is intractably large. As we do not assume access to the underlying state $\mathbf{s}$, we will deal with observations, which serve as our context $\mathbf{c}$. We are interested in representing low-dimensional contextual subspaces that approximate the high-dimensional actuations $\mathbf{u}$ of robotic agents.

This paper studies learning action interfaces that map actions $\mathbf{a} \in \mathbb{R}^k$ to raw actuation commands, $\mathbf{u} \in \mathbb{R}^n$. Throughout this work, the action space $\mathbb{A}$ will be $\mathbb{R}^k$, where $k$ is smaller than the dimensionality of the raw actuation space (e.g. number of joints) of the robotic agent. In the MDP framework, we can interpret action interfaces as being absorbed in the transition dynamics, $T(\mathbf{s}, \mathbf{a}, \mathbf{s}')$.

## 2.3 LATENT POLICY FRAMEWORK

The latent actions framework assumes that the actuation commands produced by the optimal policy $\pi^*$ exist on some lower dimensional manifold. In latent action models, latent actions $\mathbf{z} \in \mathbb{R}^k$ are mapped to this manifold. These models have typically been studied in settings where there exists a dataset of transition tuples $(\mathbf{c}, \mathbf{u}, \mathbf{c}', r)$. Here $\mathbf{c}'$ is the context observed after the agent performs actuation $\mathbf{u}$ in context $\mathbf{c}$, and $r$ is the corresponding reward. We follow this paradigm of learning from offline demonstrations, and leave the study of learning latent action models online as future work, noting that some researchers have previously studied this setting (Allshire et al., 2021).

Broadly, the class of models previously studied are conditional autoencoders. These models include a neural encoder $f_\theta(\mathbf{c}, \mathbf{u}) = \mathbf{z}$ which predicts the latent action. If the model is a variational CAE, then $f_\theta(\mathbf{c}, \mathbf{u}) = (\boldsymbol{\mu}, \boldsymbol{\sigma})$, and $\mathbf{z}$ is sampled from the Gaussian parameterized by $\boldsymbol{\mu}, \boldsymbol{\sigma}$ using the reparameterization trick (Kingma & Welling, 2014). These latent actions are then reconstructed with a decoder $g_\theta(\mathbf{z}, \mathbf{c}) = \mathbf{u}$, where $\mathbf{c}$ is assumed to contextualize how the latent action $\mathbf{z}$ should map to the higher dimensional space. In some works, there is also a latent transition model $T_\theta(\mathbf{z}, \mathbf{c}) = \mathbf{c}'$, which is trained to encourage the latent space to be predictive of transitions (Allshire et al., 2021; van der Pol et al., 2020).

The most general loss function incorporating the above models is the following:

$$\arg\min_\theta L_{\mathrm{recon}}(\mathbf{c}, \mathbf{u}, g_\theta, f_\theta) + \beta L_{\mathrm{reg}}(\mathbf{c}, \mathbf{u}, f_\theta) + \alpha L_{\mathrm{dyn}}(\mathbf{z}, \mathbf{c}, \mathbf{c}', T_\theta). \qquad (3)$$

The first term $L_{\mathrm{recon}}$ is responsible for enforcing that the reconstructed latent actions approximate the demonstration actuations. The second term, $L_{\mathrm{reg}}$ incorporates all the terms that enforce additional requirements of the latent space. The typical choice are compression terms that pack the latent codes into some desired distribution which can include the Kullback-Leibler divergence, maximum mean

discrepancy, or simply the L2-norm of $\mathbf{z}$. The third term $L_{\text{dyn}}$ is used to encourage the latent actions to be predictive of transitions. The LASER algorithm is a representative example of this framework (Allshire et al., 2021). LASER trains a latent dynamics model in conjunction with a variational autoencoder.

## 3 CONTEXTUAL SUBSPACE APPROXIMATION

In this section, we describe our proposed alternative to the conditional autoencoder paradigm of latent action models. The goal is to compute a useful map from low-dimensional task relevant actions $\mathbf{a} \in \mathbb{R}^k$ to high-dimensional actuation commands (e.g. motor torques in a robotic manipulator) $\mathbf{u} \in \mathbb{R}^n$, where $n > k$. Our approach centers on optimizing a parameterized approximation of $\mathcal{Q}^*$ (see Equation 2).

First, let us consider using a linear map from the latent space to the actuation space. Instead of a non-linear function $g_\theta : \mathbf{a}, \mathbf{c} \mapsto \mathbf{u}$ that jointly maps context vectors and actions to the actuation space, we work with a non-linear function $\mathcal{Q}_\theta : \mathbf{c} \mapsto \hat{\mathbf{Q}}$ that maps context vectors to a matrix. The matrix $\hat{\mathbf{Q}} : \mathbf{a} \mapsto \mathbf{u}$ itself is a linear map from low-dimensional actions to high dimensional actuation commands.

As $\hat{\mathbf{Q}}$ serves a similar purpose to $g_\theta$ (both map actions to actuation commands), we could consider $\hat{\mathbf{Q}}$ to be a linear decoder in the latent action framework. Then optimization of the standard reconstruction loss is formulated as follows:

$$\theta^* = \arg\min_\theta \mathbb{E}_\pi \|\mathbf{u} - \hat{\mathbf{Q}}\mathbf{a}\|_2^2 \qquad (4)$$

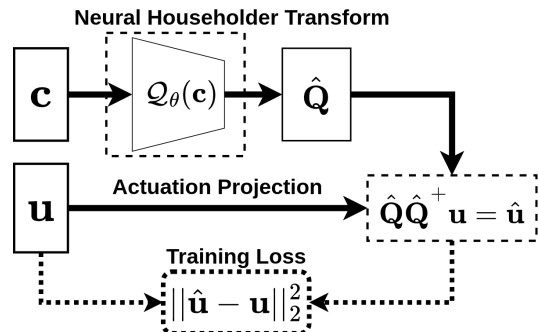

**Neural Householder Transform**

**Actuation Projection**

$\hat{\mathbf{Q}}\hat{\mathbf{Q}}^+ \mathbf{u} = \hat{\mathbf{u}}$

**Training Loss**

$\|\hat{\mathbf{u}} - \mathbf{u}\|_2^2$

Where $\hat{\mathbf{Q}} = \mathcal{Q}_\theta(\mathbf{c})$ is a function of the context $\mathbf{c}$. $\theta^*$ represents the optimal parameter vector for $\mathcal{Q}_\theta$.

The problem now becomes how to select the action $\mathbf{a} \in \mathbb{R}^k$ to use in this optimization. One approach is to follow the conditional autoencoder paradigm and predict $\mathbf{a}$ with an encoder neural network. We opt instead to compute the optimal action $\mathbf{a}^*$, which we define as the action that minimizes equation (4) for fixed $\mathbf{u}$ and $\hat{\mathbf{Q}}$ when $\theta$ is held constant:

Figure 1: Training procedure for NHT. $\mathcal{Q}_\theta$ uses a neural network and Householder transformations to map a context vector to an $n \times k$ matrix $\hat{\mathbf{Q}}$ with orthonormal columns. The data $\mathbf{u}$ associated with contex $\mathbf{c}$ is projected onto the column space of $\hat{\mathbf{Q}}$.

$$\mathbf{a}^* = \arg\min_\mathbf{a} \|\mathbf{u} - \hat{\mathbf{Q}}\mathbf{a}\|_2^2 \qquad (5)$$

Finding $\mathbf{a}^*$ is a least squares problem, and the solution can be computed with the Moore-Penrose left pseudoinverse $\hat{\mathbf{Q}}^+ = (\hat{\mathbf{Q}}^\top \hat{\mathbf{Q}})^{-1} \hat{\mathbf{Q}}^\top$. The solution to (5) is given by $\mathbf{a}^* = \hat{\mathbf{Q}}^+ \mathbf{u}$. Now our optimization problem becomes:

$$\theta^* = \arg\min_\theta \mathbb{E}\|\mathbf{u} - \hat{\mathbf{Q}}\hat{\mathbf{Q}}^+\mathbf{u}\|_2^2 \qquad (6)$$

Note that the matrix $\hat{\mathbf{Q}}\hat{\mathbf{Q}}^+$ is an orthogonal projector onto $\text{span}(\hat{\mathbf{Q}})$. Therefore, when we calculate $\hat{\mathbf{u}} = \hat{\mathbf{Q}}\hat{\mathbf{Q}}^+\mathbf{u}$, we are performing an orthogonal projection of $\mathbf{u}$ onto $\text{span}(\hat{\mathbf{Q}})$. That is, $\hat{\mathbf{u}} = \text{proj}_{\text{span}(\hat{\mathbf{Q}})}(\mathbf{u})$.

Now it is clear that the solution to Equation (6) is the best approximation attainable by $\mathcal{Q}_\theta$ to the optimal actuation subspace $W^*(\mathbf{c})$ defined in Equation (1).

### 3.1 NEURAL HOUSEHOLDER TRANSFORM

It can be desirable for the matrix $\hat{\mathbf{Q}}$ produced by $\mathcal{Q}_\theta$ to have orthonormal columns. One reason is that $\hat{\mathbf{Q}}^+$ can be trivially computed as $\hat{\mathbf{Q}}^+ = \hat{\mathbf{Q}}^\top$, which is computationally cheaper to perform. Our experimental results also indicate that learning an $\mathcal{Q}_\theta$ that produces $\hat{\mathbf{Q}}$ with orthonormal columns

tends to be more robust to hyperparameter choices (see Appendix). For these reasons we compute $\hat{\mathbf{Q}}$ by first computing an orthogonal matrix $\mathbf{Q}$, and then extracting the first $k$ columns:

$$\hat{\mathbf{Q}} = \mathbf{Q} \begin{bmatrix} \mathbf{I_k} \\ \mathbf{0} \end{bmatrix} \tag{7}$$

We can obtain $n \times n$ orthogonal matrices by computing Householder transformations. However, in order to span an arbitrary $k$-dimensional subspace, we need to chain together $k$ reflections:

$$\mathbf{Q} = \mathbf{H}(\mathbf{v}_1)\mathbf{H}(\mathbf{v}_2)\cdots\mathbf{H}(\mathbf{v}_k) \tag{8}$$

where $\mathbf{H} : \mathbb{R}^n \to \mathbb{R}^{n \times n}$ computes the Householder matrix that reflects about the hyperplane orthogonal to $\mathbf{v}_i$:

$$\mathbf{H}(\mathbf{v}_i) = \mathbf{I} - 2\mathbf{v}_i\mathbf{v}_i^\top, \quad i \in \{1,...,k\} \tag{9}$$

where each $\mathbf{v}_i$ has unit norm. Next, we describe how NHT uses a neural network to compute these $\mathbf{v}_i$ unit vectors.

### 3.1.1 Exponential Map on Unit Sphere

We would like to leverage neural networks to learn a map from contexts $\mathbf{c}$ to the $\mathbf{v}_i$ needed to compute $\hat{\mathbf{Q}}$. We can readily obtain unit $\mathbf{v}_i$ from the output of a typical neural network $h_\theta$ by simple normalization: $\mathbf{v}_i = h_\theta(\mathbf{c})/\|h_\theta(\mathbf{c})\|$. Unfortunately, this approach can result in unstable approximations. As the norm of $h_\theta(\mathbf{c})$ shrinks, arbitrarily small perturbations to the context can cause disproportionate changes in $\mathbf{v}_i$.

As a more stable alternative, we make use of the exponential map from Riemannian geometry[1] (Absil et al., 2008), which maps points in the tangent space of a manifold to the manifold itself (in our case, the sphere). We seek unit vectors in $\mathbb{R}^n$, which lie on the $(n\text{-}1)$-sphere. We can therefore make use of the exponential map on $S^{(n-1)}$ at $\mathbf{e}_1$ (the first standard basis vector, $\mathbf{e}_1 = [1, 0, \ldots, 0]^\top$). The mapping

$$\mathrm{Exp}_{\mathbf{e}_1} : \boldsymbol{\xi}_i \mapsto \mathbf{v}_i \tag{10}$$

maps[2] tangent vectors $\boldsymbol{\xi}_i \in \mathbb{R}^{(n-1)}$ to unit vectors $\mathbf{v} \in \mathbb{R}^n$. We require $k$ tangent vectors $\boldsymbol{\xi}_i$ that will map to the $\mathbf{v}_i$ vectors used to compute $\mathbf{Q}$. We therefore configure the neural network $h_\theta$ to output a vector $\bar{\boldsymbol{\xi}} \in \mathbb{R}^{k(n-1)}$. We treat the output of $h_\theta$ as $k$ stacked tangent vectors:

$$h_\theta : \mathbf{c} \mapsto \bar{\boldsymbol{\xi}}; \quad \bar{\boldsymbol{\xi}} = \left[ \boldsymbol{\xi}_1^\top, \boldsymbol{\xi}_2^\top, \ldots, \boldsymbol{\xi}_k^\top \right]^\top \tag{11}$$

We then use the exponential map on each tangent vector, resulting in a vector $\bar{\mathbf{v}} \in \mathbb{R}^{nk}$ of stacked unit $n$-vectors:

$$\begin{bmatrix} \boldsymbol{\xi}_1 \\ \boldsymbol{\xi}_2 \\ \vdots \\ \boldsymbol{\xi}_k \end{bmatrix} \xmapsto{\mathrm{Exp}} \begin{bmatrix} \mathbf{v}_1 \\ \mathbf{v}_2 \\ \vdots \\ \mathbf{v}_k \end{bmatrix} \tag{12}$$

where $\bar{\mathbf{v}} = \left[ \mathbf{v}_1^\top, \mathbf{v}_2^\top, \ldots, \mathbf{v}_k^\top \right]^\top$. Each $\mathbf{v}_i$ is then used to compute a Householder matrix (Eq. 9), which are composed to obtain $\mathbf{Q}(\bar{\mathbf{v}})$ (Eq. 8). Overall, NHT ($\mathcal{Q}_\theta : \mathbf{c} \mapsto \hat{\mathbf{Q}}$) can be understood as the composition of each of these computations:

$$\underbrace{\mathbf{c} \xmapsto{h_\theta} \bar{\boldsymbol{\xi}} \xmapsto{\mathrm{Exp}} \bar{\mathbf{v}} \xmapsto{\mathbf{Q}} \mathbf{Q}(\bar{\mathbf{v}}) \mapsto \hat{\mathbf{Q}}(\bar{\mathbf{v}})}_{\mathrm{NHT}} \tag{13}$$

where $\mathbf{c}$ is a context vector of arbitrary dimension, and $\mathbf{Q}(\bar{\mathbf{v}})$ is an $n \times n$ orthogonal matrix, and $\hat{\mathbf{Q}}(\bar{\mathbf{v}})$ is the matrix formed by the first $k$ columns of $\mathbf{Q}(\bar{\mathbf{v}})$.

---

[1]In the context of learning systems, exponential maps have been previously studied in the literature on normalizing flows (Rezende et al., 2020).

[2]For the sphere, the exponential map at $\mathbf{e}_1$ is computed as $\mathbf{v}_i = \mathbf{e}_1\cos(\|\boldsymbol{\xi}_i\|) + \frac{1}{\|\boldsymbol{\xi}_i\|}\begin{bmatrix} 0 \\ \boldsymbol{\xi}_i \end{bmatrix}\sin(\|\boldsymbol{\xi}_i\|)$.

## 3.2 EXISTENCE

If we hope to use NHT to approximate arbitrary subspaces, it is important to ensure that for every $k$-dimensional subspace $W$ of $\mathbb{R}^n$, there exists a vector $\bar{\mathbf{v}} \in \mathbb{R}^{nk}$ such that $W = \text{span}\left(\hat{\mathbf{Q}}(\bar{\mathbf{v}})\right)$.

**Remark.** *Let $W \subseteq \mathbb{R}^n$ be an arbitrary k-dimensional subspace. There is sequence of $k$ Householder reflectors $\mathbf{Q} = \mathbf{H}_1 \mathbf{H}_2 \cdots \mathbf{H}_k$ such that the first $k$ columns of $\mathbf{Q}$ are an orthonormal basis of $W$.*

*Proof.* Let $\mathbf{M}$ be an a $n \times k$ matrix whose column space is $W$. By (Trefethen & Bau, 1997) Algorithm 10.1 we can construct a QR decomposition, $\mathbf{M} = \mathbf{QR}$ where $\mathbf{Q}$ is the product of exactly $k$ Householder reflections. Now we are done because it is a basic property of QR decompositions that the first $k$ columns of $\mathbf{Q}$ are an orthonormal basis for the column space of $\mathbf{M}$, which is $W$. $\square$

Thus, given the existence of an optimal contextual subspace $W^*(\mathbf{c})$, we can be sure that there exists some $\bar{\mathbf{v}}^*$ such that $\hat{\mathbf{Q}}(\bar{\mathbf{v}}^*)$ spans $W^*$. It is left to the neural network $h_\theta$ to approximate a set of tangent vectors $\bar{\boldsymbol{\xi}}$ that map to $\bar{\mathbf{v}}^*$, given $\mathbf{c}$.

## 3.3 SMOOTHNESS OF $\mathcal{Q}_\theta$

We conjecture that low-dimensional action interfaces that change abruptly from state-to-state may degrade learning in RL agents. Thus we are interested in whether or not $\hat{\mathbf{Q}} = \mathcal{Q}_\theta(\mathbf{c})$ changes smoothly with respect to changes in the context. Concretely, we would like to limit the change in the high-dimensional actuation $\hat{\mathbf{u}}$ corresponding to an identical low-dimensional action $\mathbf{a}$ given that the change in the context is small. Let $\hat{\mathbf{Q}}_1 = \mathcal{Q}_\theta(\mathbf{c}_1)$ and $\hat{\mathbf{Q}}_2 = \mathcal{Q}_\theta(\mathbf{c}_2)$ for two nearby contexts, $\mathbf{c}_1$ and $\mathbf{c}_2$. Suppose an agent takes the same low-dimensional action $\mathbf{a}$ in both contexts. Denote the corresponding actuation commands as $\hat{\mathbf{u}}_1 = \hat{\mathbf{Q}}_1 \mathbf{a}$ and $\hat{\mathbf{u}}_2 = \hat{\mathbf{Q}}_2 \mathbf{a}$, respectively. We would like to limit the magnitude of the change in $\hat{\mathbf{u}}$ (i.e. $\|\hat{\mathbf{u}}_1 - \hat{\mathbf{u}}_2\|$) with respect to changes in context. That is, we would like to find a constant $L_u$ such that:

$$\|\hat{\mathbf{u}}_1 - \hat{\mathbf{u}}_2\| \leq L_u \|\mathbf{c}_1 - \mathbf{c}_2\| \tag{14}$$

where $\| \cdot \|$ refers to the vector 2-norm. We begin by assuming that the agent is limited to low-dimensional actions with norm less than or equal to $M$. Then we have:

$$\|\hat{\mathbf{u}}_1 - \hat{\mathbf{u}}_2\| = \|\hat{\mathbf{Q}}_1 \mathbf{a} - \hat{\mathbf{Q}}_2 \mathbf{a}\| \tag{15}$$

$$= \|(\hat{\mathbf{Q}}_1 - \hat{\mathbf{Q}}_2)\mathbf{a}\| \tag{16}$$

$$\leq \|(\hat{\mathbf{Q}}_1 - \hat{\mathbf{Q}}_2)\| \cdot \|\mathbf{a}\| \tag{17}$$

$$\leq M \|(\hat{\mathbf{Q}}_1 - \hat{\mathbf{Q}}_2)\| \tag{18}$$

where the norm in Eq. 18 is the matrix norm induced by the vector 2-norm. We now seek to limit this norm by finding a scalar constant $L$ such that

$$\|\mathcal{Q}(\mathbf{c}_1) - \mathcal{Q}(\mathbf{c}_2)\| \leq L \|\mathbf{c}_1 - \mathbf{c}_2\| \tag{19}$$

Given that such an $L$ exists, it is called a Lipschitz constant, and $\mathcal{Q}$ is considered to be $L$-Lipschitz. It turns out that there is a well understood procedure for training Lipschitz continuous neural networks Gouk et al. (2018). Using this Lipschitz regularization, we can choose a constant $L_h$ such that

$$\|\bar{\boldsymbol{\xi}}_1 - \bar{\boldsymbol{\xi}}_2\| \leq L_h \|\mathbf{c}_1 - \mathbf{c}_2\| \tag{20}$$

where $\bar{\boldsymbol{\xi}}_1 = h(\mathbf{c}_1)$ and $\bar{\boldsymbol{\xi}}_2 = h(\mathbf{c}_2)$. The exponential map on the sphere has a Lipschitz constant of 1, so we have the same result for the Lipschitz continuity of $\bar{\mathbf{v}}$ with respect to changes in context. All that remains is to show that $\mathbf{Q}(\bar{\mathbf{v}})$ is Lipschitz continuous.

**Theorem 1.** *Let $\bar{\mathbf{v}}_1, \bar{\mathbf{v}}_2 \in \mathbb{R}^{nk}$ be constructed from $k$ stacked unit $n$-vectors, and $\mathbf{Q}(\bar{\mathbf{v}})$ be the product of the corresponding Householder reflections (as defined in Eq. 8, 9). Then,*

$$\|\mathbf{Q}(\bar{\mathbf{v}}_1) - \mathbf{Q}(\bar{\mathbf{v}}_2)\| \leq L_Q \|\bar{\mathbf{v}}_1 - \bar{\mathbf{v}}_2\| \tag{21}$$

*where $L_Q = 2\sqrt{k}$.*

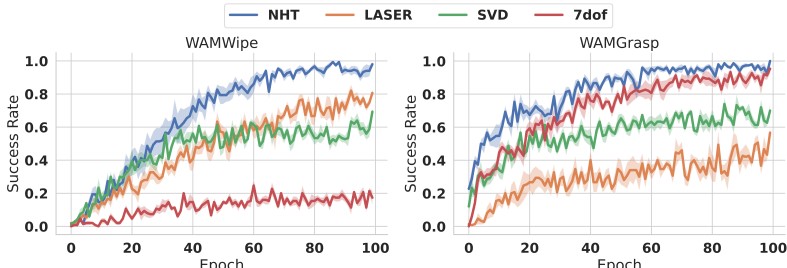

Figure 3: Learning curves corresponding to the configurations with the best average final success rate, over all hyperparameter configurations, for each method. Each curve shows the mean success rate over five runs of the best configuration, with the shaded regions indicating the standard error.

*Proof.* Please see section A.3 in the appendix. □

Using the fact that the Lipschitz constant of a composition of Lipschitz continuous functions is upper bounded by the product of the constituent Lipschitz constants Gouk et al. (2018), we combine the results of equation 20 and theorem 1 to obtain a Lipschitz constant for $\mathcal{Q}$: NHT is Lipschitz continuous with $L = 2L_h\sqrt{k}$.

Thus, the low dimensional action $\mathbf{a}$ is guaranteed to result in similar actuations in nearby contexts:

$$\|\hat{\mathbf{u}}_1 - \hat{\mathbf{u}}_2\| \leq 2L_h M\sqrt{k} \cdot \|\mathbf{c}_1 - \mathbf{c}_2\| \tag{22}$$

where $\hat{\mathbf{u}}_1 = \hat{\mathbf{Q}}_1\mathbf{a}$ and $\hat{\mathbf{u}}_2 = \hat{\mathbf{Q}}_2\mathbf{a}$, respectively.

## 4 EXPERIMENTAL SETUP

Our experimental results focus on validating the efficacy of neural householder transforms for learning kinematic tasks within a custom MuJoCo simulation of a Barrett WAM robotic manipulator with seven degrees-of-freedom (see Figure 2). We model each task as an MDP, and report results in two environments: WAMWipe and WAMGrasp. Learning involves first training an NHT model on an offline dataset of demonstrations, and then fixing the parameters of NHT and using Deep Deterministic Policy Gradient (DDPG) (Lillicrap et al., 2016) to learn a policy online. We compare DDPG agents trained with an NHT action interface against agents trained with a state-of-the-art latent action model (Allshire et al., 2021), agents trained with an actuation basis computed by SVD, and agents trained in the raw actuation space of the task (7dof joint velocity control). In our experiments we used a publicly available implementation of deep deterministic policy gradient (Andrychowicz et al., 2017)[3].

### 4.1 WAM ENVIRONMENTS

WAMWipe and WAMGrasp were designed to study the effects of using NHT to augment reinforcement learning in kinematic manipulation tasks with a binary reward function. These tasks can be classified according to how their goals are defined, and constraints on the configuration of the manipulator during execution of the task. Table 1 in the appendix enumerates the goal type and constraints present in WAMWipe and WAMGrasp. Section A.1 includes detailed descriptions of these environments.

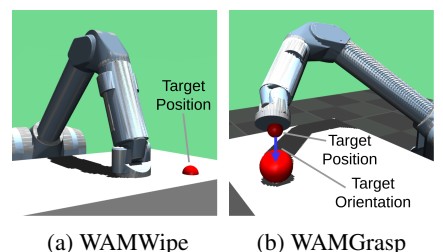

(a) WAMWipe     (b) WAMGrasp

Figure 2: Simulated kinematic manipulation environments with distinct goal types and constraints.

### 4.2 ACTION INTERFACE BASELINES

In addition to training NHT from a dataset of demonstrations, we trained LASER (Allshire et al., 2021) from the same dataset, and computed the singular value decomposition (SVD) of the dataset

---

[3]Although we used the implementation of DDPG introduced in the HER paper, we did not use HER in any of our experiments.

of joint velocities executed during the demonstrations. In our experiments, the state-conditioned actuation basis computed by NHT, static basis computed by SVD, and nonlinear decoder of LASER all serve as different choices of interface between DDPG and the raw actuation commands that determine the next state of the environment. In our WAMWipe experiments, NHT, LASER, and SVD all exposed a two-dimensional action interface to DDPG, while in WAMGrasp they all exposed three-dimensional interfaces. The $k \in \{2, 3\}$ actuation bases provided by SVD were the vectors in $\mathbb{R}^7$ corresponding to the $k$ largest singular values.

Demonstrations were collected by recording observation-actuation $(\mathbf{c}, \mathbf{u})$ pairs from PD controllers that were hand-engineered for each environment. For WAMWipe, the dataset consisted of 20,000 transitions, where a single demonstration consisted of roughly 250 transitions on average. In WAMGrasp, the dataset consisted of 100,000 transitions, where a single demonstration consisted of roughly 100 transitions on average. In both environments, the observation $\mathbf{c}$ upon which the output of NHT and LASER are conditioned was the concatenation of joint angles and Cartesian coordinates of the end-effector. LASER is regularized by the KL and dynamics terms in its loss function (see equation 3), while we regularize NHT by enforcing Lipschitz continuity with Lipschitz constant $L$ at each layer during training (Gouk et al., 2018). The Adam optimizer is used for both NHT and LASER, with learning rate $\alpha_{map}$, and otherwise default parameters. Likewise, Adam is used as the optimizer in our chosen implementation of DDPG, with learning rates $\alpha_{actor}$ and $\alpha_{critic}$ for the policy and value function, respectively.

## 4.3 HYPERPARAMETER SEARCH

We would like to estimate the performance of the best policy that could be learned by DDPG in a finite amount of time for agents trained with (1) an NHT action interface, (2) a LASER (Allshire et al., 2021) action interface, (3) an actuation basis computed by SVD, and (4) seven degree-of-freedom joint velocity actions. In addition, we would like to study the sensitivity of agent learning-dynamics to different hyperparameter configurations for each action interface. As such, we performed a random search over map (i.e. NHT, LASER) and DDPG hyperparameters. We jointly sampled 128 configurations each for NHT + DDPG,

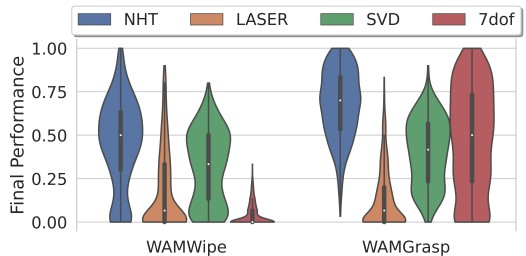

Figure 4: Violin plots of final success rates across 128 randomly sampled hyperparameter configurations (5 runs each).

LASER + DDPG, DDPG with SVD, and DDPG with joint velocity actions. For the latter two conditions the only hyperparameters of interest are those of DDPG itself. For each configuration, we first trained the mapping function (if applicable), and then trained the DDPG agent, over five runs with different random seeds. We chose to randomly sample hyperparameters because of previous work suggesting it to be more computationally efficient to find better hyper parameters (Bergstra & Bengio, 2012). The ranges and method of sampling used for each hyperparameter are listed in Table 2 of the appendix.

## 4.4 HALFCHEETAH

While our main interest for the application of NHT lies in constrained/safe robotic manipulation, there is value in validating the utility of NHT on more standard reinforcement learning environments. In addition, it is important to show that the action interface learned by NHT is useful for agents trained with various RL algorithms; not only for DDPG agents. We therefore performed a hyperparameter search experiment with a standard implementation Dhariwal et al. (2017) of PPO Schulman et al. (2017) on the HalfCheetah-v4 environment from OpenAI Gym Brockman et al. (2016). This environment has a 17-dimensional observation space that includes angular positions and velocities, and a 6-dimensional torque actuation space (compared to the joint velocity actuation space in WAMGrasp and WAMWipe). We compared NHT agents to agents that learned in the standard 6dof actuation space of HalfCheetah, and agents with LASER Allshire et al. (2021) and SVD action interfaces. NHT, SVD, and LASER all learned 2-dimensional action interfaces. This experiment precisely mirrored the hyperameter search experiments reported in section 4.3, except that

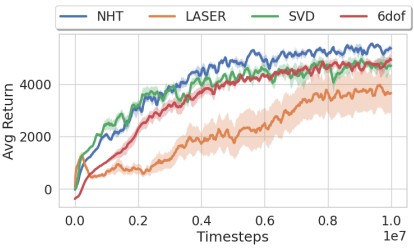 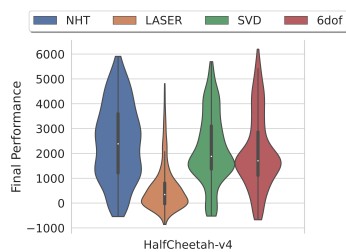

Figure 5: Results of hyperparameter search for action mapping methods in HalfCheetah-v4. **Left**: Learning curves corresponding to the configurations with the best average final success rate. **Right**: Violin plots of final success rates across 128 randomly sampled hyperparameter configurations (5 runs each).

the demonstrations used to train NHT, SVD and LASER were collected from the best-performing policy learned by the standard 6dof agent. A total of just 1,000 transitions were recorded from this expert policy. The return on this demonstration episode was over 6,000. The hyperparameter ranges and sampling methods for this experiment are summarized in Table 3 (see Appendix).

## 5 EXPERIMENTAL RESULTS

Figure 4 summarizes the results of the random hyperparameter search in the WAMWipe and WAMGrasp environments. The violin plots represent the distribution of final success rates (success rate after 100 epochs of training) across every randomly sampled hyperparameter configuration. The learning curves in Figure 3 plot the mean success rate during training for the best performing agent in each condition, averaged over five runs.

It can be seen that DDPG agents trained with an NHT action interface produced the best performing agents after hyperparameter optimization (higher success rates in fewer epochs) in both WAMWipe and WAMGrasp. In addition, the distributions of final success rates across hyperparameter configurations suggest that agents trained with NHT are more robust to hyperparameter choices compared to the baselines. Although in some runs the 7dof agent managed to reach a success rate of 100% in WAMGrasp, the variance of final success rates amongst 7dof agents is much larger than the variance of success rates for NHT agents. In general there was not a strong correlation between any one hyperparameter and the final performance of the agents (coefficient of determination $< 0.1$).

The learning curves of the agents with the best average final performance, and the distribution of final agent performances for each method in HalfCheetah-v4 are shown in Figure 5. Interestingly, we found that the constant (i.e. *not* state-dependent) action interface of SVD was sufficient to learn more efficiently than the standard 6dof agent while still achieving the same asymptotic performance. This suggests that all of the instantaneous actuations used by an expert ($> 6,000$ return) HalfCheetah agent lie close to a fixed 2-dimensional linear subspace! There appears to be some benefit to the agent learning in an adaptive actuation subspace with NHT, although the performance gains are small in this environment. The agents learning with NHT tended again to be more robust to different hyperparameter configurations.

## 6 CONCLUSION

We proposed contextual subspace approximation as a novel alternative to deep latent actions models for robotic control. We derived the Neural Householder Transform model as an approach to contextual subspace approximation, and showed that it is smooth with respect to changes in context. In a large hyperparameter search experiment, we found that reinforcement learning agents trained with NHT outperformed agents trained to act in (1) the original actuation space, (2) a global linear actuation basis computed by SVD, and (3) a state-of-the-art deep latent action model, LASER, in novel WAMWipe and WAMGrasp robotic manipulation environments, as well as in the standard HalfCheetah environment.

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

# A APPENDIX

Here we provide additional details about the WAMWipe and WAMGrasp environments, details about our hyperparameter search, and a proof for the theorem presented in Section 3.3 of the main paper. We also include in section A.4 supplementary results from an ablation study where we strip NHT of the constraint that it must output orthonormal bases. Finally, we compare NHT to a Jacobian pseudoinverse action interface in section A.5.

## A.1 RL ENVIRONMENT DETAILS

In both WAMWipe and WAMGrasp, one environment step corresponds to 10 MuJoCo simulation time-steps of length 0.002 seconds each. Both environments allow a maximum of 200 environment steps per episode.

| Env | Action Dim. | Actuation Dim. | Goal Type | Constraints |
|---|---|---|---|---|
| WAMWipe | 2 | 7 | Pos | Safety, Contact, Orientation |
| WAMGrasp | 3 | 7 | Pos, Orientation | Safety |

Table 1: Properties of reinforcement learning environments in simulation experiments.

### A.1.1 WAMWIPE

In WAMWipe the goal is to control the manipulator such that the flat face of the last link remains flush against a table while sliding to a randomly sampled goal position. The reward is -1 every step unless the end-effector is within a small distance of the goal position, in which case the reward is 0. Episode failure occurs if the end-effector: (1) Pushes into the table, (2) Lifts off of the table, or (3) The end-effector tilts such that it is no longer flush with the table. Let $\mathbf{p}$ denote the unit vector orthogonal to the face of the end-effector, pictured as a purple arrow in figure A.1. Constraint (3), the orientation constraint, was considered violated when the angle between $\mathbf{p}$ and the vector orthogonal to the surface of the table (not pictured) was greater than $\pi/16$ radians. The agent observation in our experiments was a concatenated vector of joint angles, Cartesian coordinates of the end-effector, Cartesian coordinates of the goal position, and the unit vector orthogonal to the face of the end-effector. The actions in our WAMWipe experiments were either 7dof joint velocity commands, or 2-dimensional actions input to an NHT, SVD, or LASER action interface.

### A.1.2 WAMGRASP

In WAMGrasp the goal is to simultaneously reach a randomly sampled grasp-point, while achieving a goal orientation that is determined by the grasp-point. Let $\mathbf{p}^*$ denote the unit vector pointing from the grasp-point (small sphere in Figure A.1) to the object being grasped (large sphere in Figure A.1). We consider the orientation satisfactory if the angle $\theta$ between $\mathbf{p}^*$ and the vector orthogonal to the face of the manipulator, $\mathbf{p}$, is less than $\pi/16$ radians. The reward in WAMGrasp is -1 at every step unless the end-effector is within a small distance of the grasp-point with a satisfactory orientation. Episode failure occurs if the end-effector collides with either the object being grasped (large red sphere in Figure A.1) or the table. In each episode the grasp-point is randomly sampled from the surface of a sphere with the same center but larger radius than the large red sphere in Figure A.1.

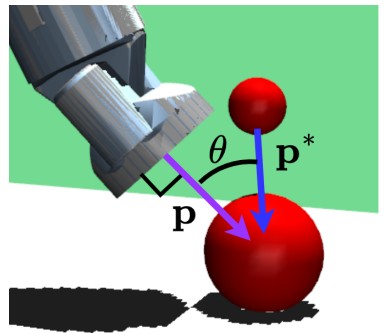

Figure A.1: The 3-vector $\mathbf{p}$ pictured here was used to determine whether the orientation constraint/goal-condition was satisfied in WAMWipe/WAMGrasp, respectively.

The agent observation was a concatenated vector of joint angles, Cartesian coordinates of the end-effector, and Cartesian coordinates of the grasp point. The actions in our WAMGrasp experiments were either 7dof joint velocity commands, or 3-dimensional actions input to an NHT, SVD, or LASER action interface.

### A.2 HYPERPARAMETER SEARCH DETAILS

The hyperparameter search experiment described in section 4.3 of the main paper was designed to estimate the performance of the best policy that could be learned by DDPG in a finite amount of time for agents trained with (1) an NHT action interface, (2) a LASER Allshire et al. (2021) action interface, (3) an actuation basis computed by SVD, and (4) seven degree-of-freedom joint velocity actions.

The hyperparameter search also enabled us to study the sensitivity of agent learning-dynamics to different hyperparameter configurations for each action interface. The results serve as empirical evidence with which to answer questions such as: "Could changing the neural architecture of NHT

cause a significant drop in the final success rate of a policy learned by DDPG?" Answering such questions is non-trivial since there may or may not be complex interactions between map (i.e. NHT, LASER) hyperparameters, DDPG hyperparameters, and final agent performance. It is unknown whether NHT hyperparameters tuned for an agent with arbitrary configuration $A$ will be the best NHT hyperparameters for an agent with a different configuration $B$. For example, it is conceivable that a DDPG agent with hyperparameter configuration $A$ may perform best with NHT configuration $C$, while DDPG with configuration $B$ performs best with NHT configuration $D$. Thus a meaningful search should jointly vary the hyperparameters of the mapping models and the DDPG agent.

We jointly sampled 128 configurations each for NHT + DDPG, LASER + DDPG, DDPG with SVD, and DDPG with joint velocity actions. For each configuration, we first trained the mapping function (if applicable), and then trained the DDPG agent, over five runs with different random seeds. The range of values and sampling method used for each hyperparameter are listed in table 2.

For both WAMWipe and WAMGrasp, each agent was trained for one million environment steps, using three workers to generate experience. This resulted in 100 training epochs of 10,000 steps each.

| Parameter | Range | Sampling |
|---|---|---|
| $\alpha_{map}$ | (1e-6, 1e-1) | Exponential |
| $\beta_{KL}$ (LASER) | (1e-6, 1e0) | Exponential |
| $\beta_{dyn}$ (LASER) | (1e-6, 1e0) | Exponential |
| $L$ (SCL) | (1e-1, 1e2) | Exponential |
| Map Batch Size | (32, 256) | Uniform |
| Map Activation | {ReLU, sigmoid, tanh} | Uniform |
| Map Hidden Layers | {1, 2, 3, 4} | Uniform |
| Map Hidden Units | (16, 1024) | Geometric |
| $\alpha_{actor}$ | (1e-4, 1e-2) | Exponential |
| $\alpha_{critic}$ | (1e-4, 1e-2) | Exponential |
| Rand Action $\epsilon$ | (0, 0.4) | Uniform |
| Action Noise $\sigma$ | (0, 0.4) | Uniform |
| Penalty on $\|\mathbf{a}\|$ | (0, 1) | Uniform |
| Max $\|\mathbf{a}\|$ | (1, 10)* | Exponential |
| DDPG Batch Size | (32, 256) | Uniform |
| Polyak | (0.9, 0.99) | Uniform |

Table 2: Hyperparameter ranges and sampling methods for pre-trained mapping functions (top) and DDPG (bottom). We use 1e$x$ as shorthand to denote $1 \times 10^{x}$. Exponential and Geometric sampling of parameters was carried out as described in Bergstra & Bengio (2012). *For the agent with 7dof actions we use the range (0.1, 10).

The parameter ranges and sampling methods for the PPO agent used in the HalfCheetah hyperparameter search are given in table 3.

| Parameter | Range | Sampling |
|---|---|---|
| $\alpha$ | (1e-4, 1e-2) | Exponential |
| Discount Factor $\gamma$ | (0.9, 0.99) | Uniform |
| GAE Parameter $\lambda$ | (0.9, 0.99) | Uniform |
| VF coefficient | (0.25,1) | Uniform |
| Clipping Parameter $\epsilon$ | (0.05,0.5) | Uniform |
| Steps-per-update | {1024,2048,4096} | Uniform |

Table 3: Hyperparameter ranges for PPO in the HalfCheetah-v4 experiments. We use 1e$x$ as shorthand to denote $1 \times 10^{x}$. Exponential sampling of parameters was carried out as described in Bergstra & Bengio (2012).

### A.3 SMOOTHNESS OF $\mathbf{Q}(\bar{\mathbf{v}})$

In this section we prove the Lipschitz continuity of $\mathbf{Q}(\bar{\mathbf{v}})$, as stated in theorem 1.

**Theorem 1.** *Let $\bar{\mathbf{v}}_1, \bar{\mathbf{v}}_2 \in \mathbb{R}^{nk}$ be constructed from $k$ stacked unit $n$-vectors, and $\mathbf{Q}(\bar{\mathbf{v}})$ be the product of the corresponding Householder reflections (as defined in Eq. 8, 9). Then,*

$$\|\mathbf{Q}(\bar{\mathbf{v}}_1) - \mathbf{Q}(\bar{\mathbf{v}}_2)\| \le L_Q \|\bar{\mathbf{v}}_1 - \bar{\mathbf{v}}_2\| \tag{21}$$

*where $L_Q = 2\sqrt{k}$.*

We write $\mathbf{H}_i$ as shorthand for $\mathbf{H}(\mathbf{v}_i) = \mathbf{I} - 2\mathbf{v}_i\mathbf{v}_i^\top$. We write $\bar{\mathbf{v}} \in \mathbb{R}^{nk}$ to denote the concatenated column vector of $\mathbf{v}_i \in \mathbb{R}^n$: $\bar{\mathbf{v}} = [\mathbf{v}_1^\top, \mathbf{v}_2^\top, \dots, \mathbf{v}_k^\top]^\top$. We denote the map from $\bar{\mathbf{v}}$ to the corresponding product of reflections as $Q : \bar{\mathbf{v}} \mapsto \mathbf{Q}(\bar{\mathbf{v}})$, where

$$\mathbf{Q}(\bar{\mathbf{v}}) = \mathbf{H}(\mathbf{v}_1)\mathbf{H}(\mathbf{v}_2)\cdots\mathbf{H}(\mathbf{v}_k) \tag{23}$$

We likewise write $\bar{\boldsymbol{\delta}} \in \mathbb{R}^{nk}$ to denote the concatenated vector of perturbations to each $\mathbf{v}_i$

$$\bar{\boldsymbol{\delta}} = \begin{bmatrix} \boldsymbol{\delta}_1' \\ \boldsymbol{\delta}_2' \\ \vdots \\ \boldsymbol{\delta}_k' \end{bmatrix} = \begin{bmatrix} c_1\boldsymbol{\delta}_1 \\ c_2\boldsymbol{\delta}_2 \\ \vdots \\ c_k\boldsymbol{\delta}_k \end{bmatrix} \tag{24}$$

where $\|\bar{\boldsymbol{\delta}}\| = 1$, with scalars $c_i \in \mathbb{R}$ scaling the unit norm $\boldsymbol{\delta}_i$ vectors that represent the direction of change for each $\mathbf{v}_i$. We consider the directional derivative of $\mathbf{Q}(\bar{\mathbf{v}})$ in the direction of $\bar{\boldsymbol{\delta}}$:

$$\nabla_{\bar{\boldsymbol{\delta}}}\mathbf{Q}(\bar{\mathbf{v}}) \doteq \lim_{\epsilon \to 0} \frac{\mathbf{Q}(\bar{\mathbf{v}} + \epsilon\bar{\boldsymbol{\delta}}) - \mathbf{Q}(\bar{\mathbf{v}})}{\epsilon} \tag{25}$$

where $\|\bar{\boldsymbol{\delta}}\| = 1$.

The existence of a positive constant $L_Q$ that bounds $\|\nabla_{\bar{\boldsymbol{\delta}}}\mathbf{Q}(\bar{\mathbf{v}})\|$ implies Lipschitz continuity of $\mathbf{Q}(\bar{\mathbf{v}})$:

$$\|\mathbf{Q}(\bar{\mathbf{v}}_1) - \mathbf{Q}(\bar{\mathbf{v}}_2)\| \le L_Q \|\bar{\mathbf{v}}_1 - \bar{\mathbf{v}}_2\| \tag{26}$$

for all $\bar{\mathbf{v}}_1, \bar{\mathbf{v}}_2$ constructed with $k$ stacked unit $n$-vectors. We explicitly compute such an $L_Q$ below. As a first step, we show in section A.3.1 that $\|\nabla_{\boldsymbol{\delta}}\mathbf{H}(\mathbf{v})\| = 2$. We will then use this result to compute an upper bound on $L_Q$ in section A.3.2.

### A.3.1 LIPSCHITZ CONTINUITY OF $\mathbf{H}(\mathbf{v})$

The directional derivative of $\mathbf{H}(\mathbf{v})$ in the direction of $\boldsymbol{\delta}$ is defined as:

$$\nabla_{\boldsymbol{\delta}}\mathbf{H}(\mathbf{v}) \doteq \lim_{\epsilon \to 0} \frac{\mathbf{H}(\mathbf{v} + \epsilon\boldsymbol{\delta}) - \mathbf{H}(\mathbf{v})}{\epsilon} \tag{27}$$

where $\boldsymbol{\delta} \in \mathbb{R}^n$. Recall that $\mathbf{v}$ is in the $n-1$ sphere, and thus any instantaneous change to $\mathbf{v}$ must occur in a direction tangent to the sphere at $\mathbf{v}$; that is, $\boldsymbol{\delta} \perp \mathbf{v}$. Furthermore, without loss of generality we let $\|\boldsymbol{\delta}\| = 1$. Thus, $\boldsymbol{\delta}$ is a unit vector in the direction of the perturbation of $\mathbf{v}$.

We first simplify the first term in the numerator:

$$\mathbf{H}(\mathbf{v} + \epsilon\boldsymbol{\delta}) = \mathbf{I} - 2(\mathbf{v} + \epsilon\boldsymbol{\delta})(\mathbf{v} + \epsilon\boldsymbol{\delta})^\top \tag{28}$$

$$= \mathbf{I} - 2(\mathbf{v}\mathbf{v}^\top + \epsilon\boldsymbol{\delta}\mathbf{v}^\top + \epsilon\mathbf{v}\boldsymbol{\delta}^\top + \epsilon^2\boldsymbol{\delta}\boldsymbol{\delta}^\top) \tag{29}$$

$$= \mathbf{I} - 2\mathbf{v}\mathbf{v}^\top - 2(\epsilon\boldsymbol{\delta}\mathbf{v}^\top + \epsilon\mathbf{v}\boldsymbol{\delta}^\top + \epsilon^2\boldsymbol{\delta}\boldsymbol{\delta}^\top) \tag{30}$$

$$= \mathbf{H}(\mathbf{v}) - 2(\epsilon\boldsymbol{\delta}\mathbf{v}^\top + \epsilon\mathbf{v}\boldsymbol{\delta}^\top + \epsilon^2\boldsymbol{\delta}\boldsymbol{\delta}^\top) \tag{31}$$

Substituting the result into the definition of $\nabla_{\boldsymbol{\delta}}\mathbf{H}(\mathbf{v})$, we have:

$$\nabla_{\boldsymbol{\delta}}\mathbf{H}(\mathbf{v}) = \lim_{\epsilon \to 0} \frac{-2(\epsilon\boldsymbol{\delta}\mathbf{v}^\top + \epsilon\mathbf{v}\boldsymbol{\delta}^\top + \epsilon^2\boldsymbol{\delta}\boldsymbol{\delta}^\top)}{\epsilon} \tag{32}$$

$$= \lim_{\epsilon \to 0} -2(\boldsymbol{\delta}\mathbf{v}^\top + \mathbf{v}\boldsymbol{\delta}^\top + \epsilon\boldsymbol{\delta}\boldsymbol{\delta}^\top) \tag{33}$$

$$= -2(\boldsymbol{\delta}\mathbf{v}^\top + \mathbf{v}\boldsymbol{\delta}^\top) \tag{34}$$

Now we compute $\|\nabla_{\boldsymbol{\delta}}\mathbf{H}(\mathbf{v})\|$. Note that the symmetry of the sphere guarantees that $\|\nabla_{\boldsymbol{\delta}}\mathbf{H}(\mathbf{v})\|$ is invariant with respect to both $\boldsymbol{\delta}$ and $\mathbf{v}$.

$$\|\nabla_{\boldsymbol{\delta}}\mathbf{H}(\mathbf{v})\| \doteq \max_{\mathbf{x}\neq 0} \frac{\|\nabla_{\boldsymbol{\delta}}\mathbf{H}(\mathbf{v})\mathbf{x}\|}{\|\mathbf{x}\|} \tag{35}$$

The numerator is maximized when $\mathbf{x}$ is in the plane spanned by $\mathbf{v}$ and $\boldsymbol{\delta}$. Given this is the case, we can write $\mathbf{x}$ as a linear combination of $\mathbf{v}$ and $\boldsymbol{\delta}$. Let

$$\mathbf{x} = \alpha\mathbf{v} + \beta\boldsymbol{\delta} \tag{36}$$

for some $\alpha, \beta \in \mathbb{R}$. We then have the following:

$$\|\nabla_{\boldsymbol{\delta}}\mathbf{H}(\mathbf{v})\| = \max_{\mathbf{x}\neq 0} \frac{\|\nabla_{\boldsymbol{\delta}}\mathbf{H}(\mathbf{v})\mathbf{x}\|}{\|\mathbf{x}\|} \tag{37}$$

$$= \max_{\mathbf{x}\neq 0} \frac{\| -2(\boldsymbol{\delta}\mathbf{v}^\top + \mathbf{v}\boldsymbol{\delta}^\top)\mathbf{x}\|}{\|\mathbf{x}\|} \tag{38}$$

$$= 2\frac{\|(\boldsymbol{\delta}\mathbf{v}^\top + \mathbf{v}\boldsymbol{\delta}^\top)(\alpha\mathbf{v} + \beta\boldsymbol{\delta})\|}{\|\mathbf{x}\|} \tag{39}$$

$$= 2\frac{\|\alpha\boldsymbol{\delta}\mathbf{v}^\top\mathbf{v} + \alpha\mathbf{v}\boldsymbol{\delta}^\top\mathbf{v} + \beta\boldsymbol{\delta}\mathbf{v}^\top\boldsymbol{\delta} + \beta\mathbf{v}\boldsymbol{\delta}^\top\boldsymbol{\delta}\|}{\|\mathbf{x}\|} \tag{40}$$

$$= 2\frac{\|\alpha\boldsymbol{\delta}\mathbf{v}^\top\mathbf{v} + \beta\mathbf{v}\boldsymbol{\delta}^\top\boldsymbol{\delta}\|}{\|\mathbf{x}\|} \tag{41}$$

$$= 2\frac{\|\alpha\boldsymbol{\delta} + \beta\mathbf{v}\|}{\|\mathbf{x}\|} \tag{42}$$

where equation 41 follows from 40 by the fact that $\boldsymbol{\delta} \perp \mathbf{v}$. Equation 42 follows from the fact that both $\boldsymbol{\delta}$ and $\mathbf{v}$ have unit norm.

Now, recall that $\mathbf{x} = \alpha\mathbf{v} + \beta\boldsymbol{\delta}$. The numerator in 42 represents a simple change of basis for $\mathbf{x}$. Since $\boldsymbol{\delta}$ and $\mathbf{v}$ are orthonormal, this change of basis preserves the norm of $\mathbf{x}$. Hence $\|\alpha\boldsymbol{\delta} + \beta\mathbf{v}\| = \|\mathbf{x}\|$, and we have:

$$\|\nabla_{\boldsymbol{\delta}}\mathbf{H}(\mathbf{v})\| = 2 \tag{43}$$

This implies $\mathbf{H}(\mathbf{v})$ is Lipschitz continuous with Lipschitz constant 2.

### A.3.2   LIPSCHITZ CONTINUITY OF $\mathbf{Q}$

We now consider the directional derivative of $\mathbf{Q}(\bar{\mathbf{v}})$ in the direction of $\bar{\boldsymbol{\delta}}$:

$$\nabla_{\bar{\boldsymbol{\delta}}}\mathbf{Q}(\bar{\mathbf{v}}) \doteq \lim_{\epsilon\to 0} \frac{\mathbf{Q}(\bar{\mathbf{v}} + \epsilon\bar{\boldsymbol{\delta}}) - \mathbf{Q}(\bar{\mathbf{v}})}{\epsilon} \tag{44}$$

where $\|\bar{\boldsymbol{\delta}}\| = 1$. Recall:

$$\bar{\boldsymbol{\delta}} = \begin{bmatrix} \boldsymbol{\delta}_1' \\ \boldsymbol{\delta}_2' \\ \vdots \\ \boldsymbol{\delta}_k' \end{bmatrix} = \begin{bmatrix} c_1\boldsymbol{\delta}_1 \\ c_2\boldsymbol{\delta}_2 \\ \vdots \\ c_k\boldsymbol{\delta}_k \end{bmatrix} \tag{45}$$

**Theorem 1.** *Let $\bar{\mathbf{v}}_1, \bar{\mathbf{v}}_2 \in \mathbb{R}^{nk}$ be constructed from $k$ stacked unit $n$-vectors, and $\mathbf{Q}(\bar{\mathbf{v}})$ be the product of the corresponding Householder reflections (as defined in Eq. 8, 9). Then,*

$$\|\mathbf{Q}(\bar{\mathbf{v}}_1) - \mathbf{Q}(\bar{\mathbf{v}}_2)\| \leq L_Q \|\bar{\mathbf{v}}_1 - \bar{\mathbf{v}}_2\| \tag{21}$$

*where $L_Q = 2\sqrt{k}$.*

*Proof.* We begin by expanding the numerator of $\nabla_{\bar{\boldsymbol{\delta}}}\mathbf{Q}$

$$\nabla_{\bar{\boldsymbol{\delta}}}\mathbf{Q}(\bar{\mathbf{v}}) = \lim_{\epsilon\to 0} \frac{\mathbf{H}(\mathbf{v}_1 + \epsilon\boldsymbol{\delta}_1')\mathbf{H}(\mathbf{v}_2 + \epsilon\boldsymbol{\delta}_2')\cdots\mathbf{H}(\mathbf{v}_k + \epsilon\boldsymbol{\delta}_k') - \mathbf{Q}(\bar{\mathbf{v}})}{\epsilon} \tag{46}$$

We now consider the first term in the numerator. In the following we write $\mathbf{H}_i$ as shorthand for $\mathbf{H}(\mathbf{v}_i)$, and $\nabla \mathbf{H}_i$ as shorthand for $\nabla_{\boldsymbol{\delta}_i} \mathbf{H}(\mathbf{v}_i)$, the derivative of $\mathbf{H}(\mathbf{v}_i)$ as defined in equation (27).

$$\mathbf{H}(\mathbf{v}_1 + \epsilon \boldsymbol{\delta}_1') \mathbf{H}(\mathbf{v}_2 + \epsilon \boldsymbol{\delta}_2') \cdots \mathbf{H}(\mathbf{v}_k + \epsilon \boldsymbol{\delta}_k') \tag{47}$$

$$= (\mathbf{H}_1 + \epsilon c_1 \nabla \mathbf{H}_1)(\mathbf{H}_2 + \epsilon c_2 \nabla \mathbf{H}_2) \cdots (\mathbf{H}_k + \epsilon c_k \nabla \mathbf{H}_k) + O(\epsilon^2) \tag{48}$$

$$= \mathbf{H}_1 \mathbf{H}_2 \cdots \mathbf{H}_k + \epsilon c_1 (\nabla \mathbf{H}_1) \mathbf{H}_2 \cdots \mathbf{H}_k + \epsilon c_2 \mathbf{H}_1 (\nabla \mathbf{H}_2) \mathbf{H}_3 \cdots \mathbf{H}_k + \dots \tag{49}$$

$$+ \epsilon c_k \mathbf{H}_1 \mathbf{H}_2 \cdots \mathbf{H}_{k-1} (\nabla \mathbf{H}_k) + O(\epsilon^2) \tag{50}$$

$$= \mathbf{Q}(\bar{\mathbf{v}}) + \epsilon c_1 (\nabla \mathbf{H}_1)(\prod_{i=2}^{k} \mathbf{H}_i) + \epsilon c_2 \mathbf{H}_1 (\nabla \mathbf{H}_2)(\prod_{i=3}^{k} \mathbf{H}_i) + \cdots + \epsilon c_k (\prod_{i=1}^{k-1} \mathbf{H}_i)(\nabla \mathbf{H}_k) + O(\epsilon^2) \tag{51}$$

$$= \mathbf{Q}(\bar{\mathbf{v}}) + \epsilon \sum_{j=1}^{k} c_j \left[ (\prod_{i=1}^{j-1} \mathbf{H}_i)(\nabla \mathbf{H}_j)(\prod_{l=j+1}^{k} \mathbf{H}_l) \right] + O(\epsilon^2) \tag{52}$$

and substitute the result into the definition of $\nabla_{\bar{\boldsymbol{\delta}}} \mathbf{Q}(\bar{\mathbf{v}})$:

$$\nabla_{\bar{\boldsymbol{\delta}}} \mathbf{Q}(\bar{\mathbf{v}}) = \lim_{\epsilon \to 0} \frac{\mathbf{Q}(\bar{\mathbf{v}}) + \epsilon \sum_{j=1}^{k} c_j \left[ (\prod_{i=1}^{j-1} \mathbf{H}_i)(\nabla \mathbf{H}_j)(\prod_{l=j+1}^{k} \mathbf{H}_l) \right] + O(\epsilon^2) - \mathbf{Q}(\bar{\mathbf{v}})}{\epsilon} \tag{53}$$

$$= \sum_{j=1}^{k} c_j \left[ (\prod_{i=1}^{j-1} \mathbf{H}_i)(\nabla \mathbf{H}_j)(\prod_{l=j+1}^{k} \mathbf{H}_l) \right] \tag{54}$$

We now work toward bounding the norm of $\nabla_{\bar{\boldsymbol{\delta}}} \mathbf{Q}(\bar{\mathbf{v}})$:

$$\|\nabla_{\bar{\boldsymbol{\delta}}} \mathbf{Q}(\bar{\mathbf{v}})\| = \left\| \sum_{j=1}^{k} c_j \left[ (\prod_{i=1}^{j-1} \mathbf{H}_i)(\nabla \mathbf{H}_j)(\prod_{l=j+1}^{k} \mathbf{H}_l) \right] \right\| \tag{55}$$

$$\leq \sum_{j=1}^{k} c_j \left\| (\prod_{i=1}^{j-1} \mathbf{H}_i)(\nabla \mathbf{H}_j)(\prod_{l=j+1}^{k} \mathbf{H}_l) \right\| \tag{56}$$

$$= \sum_{j=1}^{k} c_j \|\nabla \mathbf{H}_j\| \tag{57}$$

$$\leq \left( \sqrt{\sum_{j=1}^{k} c_j^2} \right) \left( \sqrt{\sum_{j=1}^{k} \|\nabla \mathbf{H}_j\|^2} \right) \tag{58}$$

$$= \|\bar{\boldsymbol{\delta}}\| \sqrt{\sum_{j=1}^{k} \|\nabla_{\boldsymbol{\delta}_j} \mathbf{H}(\mathbf{v}_j)\|^2} \tag{59}$$

$$= \sqrt{\sum_{j=1}^{k} (2)^2} \tag{60}$$

$$= 2\sqrt{k} \tag{61}$$

Where equation (57) is thanks to the fact that each of the $\mathbf{H}_i$ in the preceding equation are orthogonal, and equation (58) follows by the Cauchy-Schwarz inequality.

Hence, the norm of the directional derivative of $\mathbf{Q}(\bar{\mathbf{v}})$ is bounded by $2\sqrt{k}$; that is:

$$\|\nabla \mathbf{Q}(\bar{\mathbf{v}})\| \leq 2\sqrt{k} \tag{62}$$

which implies

$$\|\mathbf{Q}(\bar{\mathbf{v}}_1) - \mathbf{Q}(\bar{\mathbf{v}}_2)\| \leq L_Q \|\bar{\mathbf{v}}_1 - \bar{\mathbf{v}}_2\| \tag{63}$$

with Lipschitz constant $L_Q = 2\sqrt{k}$. $\qquad \square$

A.4    ABLATION OF ORTHONORMAL CONSTRAINT

What is the benefit of enforcing orthonormal actuation bases in NHT? Beside the fact that the pseudoinverse of $\hat{\mathbf{Q}}$ can be computed trivially as the transpose during training, we wanted to find out if there was any empirical benefit. To answer this question we performed an experiment in which we trained a neural network to produce an arbitrary state-conditioned matrix as an actuation basis for WAMWipe and WAMGrasp. Unlike NHT, this baseline is not constrained to output a matrix with orthonormal columns. We will refer to the baseline as the state-conditioned linear map (SCL) model. The SCL baseline is a neural network $h_\theta : \mathbf{c} \mapsto \hat{\mathbf{B}} \in \mathbb{R}^{n \times k}$ that maps context vectors to an $n \times k$ matrix $\hat{\mathbf{B}}$. In our WAMWipe experiment $n = 7$ and $k = 2$, while for WAMGrasp $n = 7$ and $k = 3$.

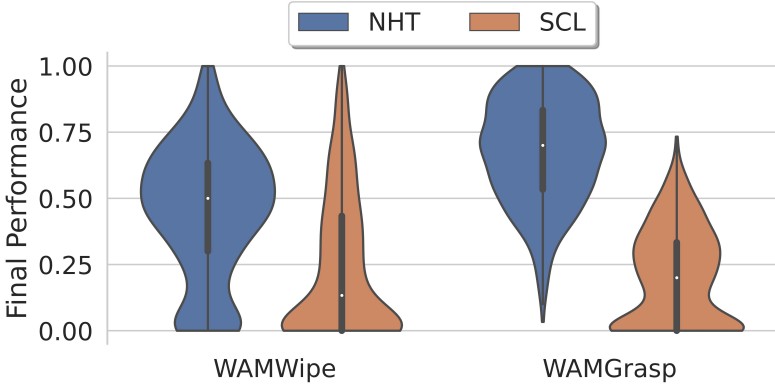

Figure A.2: Violin plots of final success rates across 128 randomly sampled hyperparameter configurations (5 runs each) for NHT vs a non-orthonormal state-conditioned linear (SCL) action mapping baseline.

As in section 4.3 of the main text, we performed a hyperparameter search in which we sampled 128 hyperparameter configurations and trained NHT/SCL and then a DDPG agent with 5 different random seeds. The sampling method and ranges for each parameter were the same for both NHT and SCL, and are listed in Table 2 of section A.2. Figure A.2 summarizes the results of this hyperparameter search with violin plots of the final performance attained by agents with NHT and SCL action interfaces for the WAMWipe and WAMGrasp environments. The NHT hyperparameter search results reported in this figure are the same as those reported in section 4.3 of the main text.

We found that 64 out of 640 (10%) of the runs for SCL in WAMWipe failed due to numerical instability. In these cases the matrices output by the unconstrained neural network had large norms, resulting in very large joint velocity actuations that caused the mujoco simulations to fail. Interestingly, we did not observe the same numerical stability issues in the SCL models that were trained for WAMGrasp. Note that, in contrast, for NHT numerical stability is not an empirical issue. The 2-norm of the matrix produced by NHT is guaranteed to be equal to one.

In WAMWipe, the best hyperparameter configurations of SCL resulted in actuation interfaces that were suitable for the DDPG agent to achieve 100% success rate. However, in both WAMWipe and WAMGrasp, the distributions of final agent performance in figure A.2 indicate that NHT was more robust than SCL with respect to variation in hyperparameter configurations. This suggests that NHT may be less sensitive to different choices of hyperparameters, making it easier to tune in practice.

A.5    COMPARISON TO JACOBIAN PSEUDOINVERSE INTERFACE

Here we compare NHT to an additional choice of action interface that, unlike the baselines discussed in the main text, is not learned from demonstrations. The Jacobian of the robotic manipulator describes the relationship between the joint velocities and the Cartesian and angular velocity of the end-effector. The pseudoinverse of the Jacobian can be used to define a six-dimensional action interface for an RL agent: 3 dimensions in the agent's action space correspond to Cartesian velocity, and the remaining 3 correspond to angular velocity.

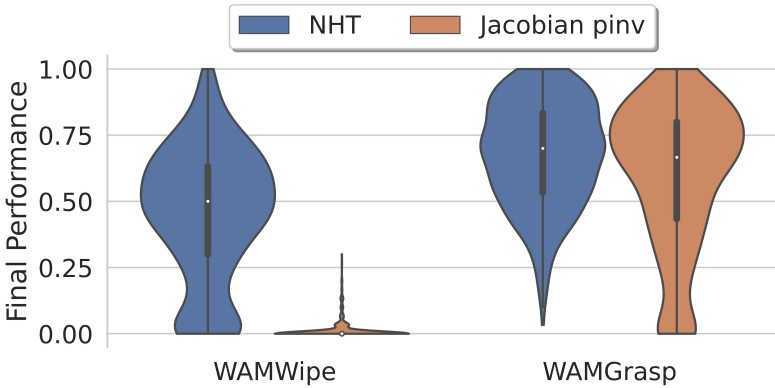

Figure A.3: Violin plots of final success rates across 128 randomly sampled hyperparameter configurations (5 runs each) for NHT vs a Jacobian pseudoinverse (Jacobian pinv) action mapping baseline.

We compared NHT to the Jacobian pseudoinverse as an action interface for a DDPG agent in WAMGrasp and WAMWipe in a hyperparameter search experiment with the same methodology described in section 4.3 of the main text (128 hyperparameter configurations, 5 seeds for each configuration). As already noted, the dimensionality of the agent's action space was 6 when using the Jacobian pseudoinverse interface. NHT was used to learn a 2-dimensional action interface for WAMWipe, and a 3-dimensional action interface for WAMGrasp. The hyperparameter search results are plotted in figure A.3. The NHT hyperparameter search results reported in this figure are the same as those reported in section 4.3 of the main text. The variation in performance for the Jacobian pseudoinverse agents are entirely due to different DDPG agent configurations (the Jacobian has no hyperparameters).

As expected, the agent with the Jacobian pseudoinverse action interface performed poorly in WAMWipe; like the 7dof joint velocity agent, the Jacobian pseudoinverse agent was able to freely jam the end-effector of the robot into the table, or lift the end-effector from the table, resulting in an automatic failure for its training episodes. Without a learned action interface, the exploratory behavior inherent in reinforcement learning resulted in destructive behavior that made learning in the highly constrained environment of WAMWipe difficult.

In WAMGrasp, the Jacobian pseudoinverse agents were sometimes able to learn to achieve 100% success rate. However, it is clear from the violin plots in figure A.3 that limiting the joint velocity commands to useful subspaces learned by NHT has some benefit over allowing free exploration with the Jacobian pseudoinverse interface. Some of the poorer hyperparameter configurations resulted in close to 0% success rate when interacting with WAMGrasp through the Jacobian pseudoinverse interface. NHT tended to concentrate agent performance, over all hyperparameter configurations, toward a success rate of 75% to 100%.

