# OpenReview forum: "Contextual Subspace Approximation with Neural Householder Transforms"
_ICLR.cc/2023/Conference — Submitted to ICLR 2023_

### Official Review · Reviewer_2zLC · 2022-10-24

**Confidence:** 4
**Correctness:** 4
**Technical Novelty And Significance:** 3
**Empirical Novelty And Significance:** 3
**Recommendation:** 5

**Clarity, Quality, Novelty And Reproducibility:**

The paper is unclear at times about the NHT and the author's contribution, please see above. Otherwise the paper is quite well written and the method is well introduced.

Some minor comments:
* In the introduction: is the last paragraph meant specifically for NHT?
* Section 2.3: in some works -> please cite
* How do you fix k? Can it be estimated/learned? Any works that do so?
* Is section 3.1.1. part of the contribution?
* In Figure 3, what happens after 100 epochs? Does the alternative approach catch up?
* How do you decide what is the right observation/context for the environments? (e.g. WAM wipe and grasp)
* Why would/should agents trained with NHT be more robust w.r.t. different hyperparameter configurations?

**Strength And Weaknesses:**

The paper is well written and the introduced approach is quite nice in my opinion.

Strengths:

* The Neural Householder Transform and the way it is computed, can yield a flexible way to compute locally well approximating linear subspaces, as indeed shown by the paper.

* By bounding the network with a Lipschitz bound, the authors can also show that the computed transform is smooth. This result can perhaps be used computationally as well, however the authors have not done so (e.g. during training).

Weaknesses:

* The claims of the paper about the contributions are not rigorously demonstrated, the authors themselves use "tend to", to describe the effect of the framework (when compared with competing approaches). This can be remedied by either more detailed experiments, and/or by a more stringent analysis of the failures/benefits of the approach.

* It is not clear if some of the NHT is the author's contribution, e.g. the particular output representation utilizing the exponential map.

**Summary Of The Paper:**

This paper presents a new latent action model based on Neural Householder Transform, where context dependent linear actuation subspaces are the outputs of an encoder-like neural network. The encoder outputs can be transformed via the exponential map on the sphere to form Household reflectors, that can be used to compute the context-dependent subspace. These local subspaces when interfaced with standard RL approaches can lead to more efficient and robust (w.r.t hyperparameters) learning in some environments.

**Summary Of The Review:**

I would recommend the acceptance of the paper, see above.

=== POST-REBUTTAL EDIT ===

Thanks to the authors for providing a careful rebuttal. As mentioned before, the method proposed in the paper is quite nice and novel in my opinion. However upon deeper reading the paper, the comments and the rebuttal, I am not convinced about the results of the paper. In particular relating to the hyperparameter-robustness, I am not convinced of the link between the NHT transform and increased robustness to hyperparameters. It's not clear to me why the context dependent linear subspaces emitted by the NHT transform would lead to such robustness, as opposed to the nonlinear dimensionality reduction achieved by LASER. I would rather expect improved learning curves, and better generalization (as I noted in my reply to the authors).

As hyperparameter-robustness seems to be the main claim of the paper (as an improvement), I'm lowering my score to 5, but I'm happy to reconsider if the reviewers would like to give some feedback on my concerns.

---

> ### Author Response · Authors · 2022-11-12
> **Initial Response to Reviewer 2zLC**
>
> We thank reviewer 2zLC for their insight, and positive comments on our work. We leave itemized responses to the concerns cited by reviewer 2zLC below. If reviewer 2zLC has any additional comments, we will provide further clarifications.

---

> > ### Author Response · Authors · 2022-11-12
> > **Claims, Experimental Design**
> >
> > **Comment:** The claims of the paper about the contributions are not rigorously demonstrated, the authors themselves use "tend to", to describe the effect of the framework (when compared with competing approaches). This can be remedied by either more detailed experiments, and/or by a more stringent analysis of the failures/benefits of the approach.
> >
> > **Response:**
> >
> > We have replaced the phrase “tend to” with more appropriate terminology such as “our results suggest”. We chose these terms particularly because of the issue of reproducibility in deep RL literature, as shown in previous work e.g. Henderson et. al., 2018. To the reviewers point, this is precisely why we performed extensive experiments choosing to compare across 128 configurations of hyperparameters for each model, with 5 runs (seeds) of each configuration, for a total of 640 runs per model in each environment. The learning curve results represent the best performing hyperparameter configurations for each method. We also included ablation results comparing our choice of orthonormal representation to non-orthonormal representation, as well as an alternative Jacobian pseudoinverse approach specific for our robotic experiments (see Appendix sections A.4 and A.5). Please let us know if there are any specific concerns related to our experimental design.
> >
> > [1] Peter Henderson, Riashat Islam, Philip Bachman, Joelle Pineau, Doina Precup, David Meger. Deep Reinforcement Learning that Matters. AAAI, 2018.

---

> > ### Author Response · Authors · 2022-11-12
> > **Contributions: Exponential map in NHT**
> >
> > **Comment:** It is not clear if some of the NHT is the author's contribution, e.g. the particular output representation utilizing the exponential map.
> >
> > **Response:**
> >
> > The representation of high-dimensional actuations as linear combinations of context-dependent basis vectors is one contribution of this work (i.e. contextual subspace approximation). NHT is an approach to learn this representation, and NHT itself (i.e. the model/algorithm) is a novel contribution. Each constituent component of NHT has existed independently for some time, e.g. the exponential map comes from Riemannian geometry, which has been developed since the 19th century. Arranging these smooth functions (Lipschitz regularized neural network -> exponential map -> householder reflection) as NHT, and using this for contextual subspace approximation is the contribution of our work. Proving the Lipschitz continuity of NHT is an additional contribution.

---

> > ### Author Response · Authors · 2022-11-12
> > **Clarification in Introduction**
> >
> > **Comment:** In the introduction: is the last paragraph meant specifically for NHT?
> >
> > **Response:**
> > If the reviewer is referring to the paragraph starting with “Additional work has…”, this is meant to contextualize the related work on where householder matrices have been studied outside of our own research. We have now combined this paragraph with the previous paragraph to avoid confusion.

---

> > ### Author Response · Authors · 2022-11-12
> > **Missing Citations**
> >
> > **Comment:** Section 2.3: in some works -> please cite
> >
> > **Response:**
> > We have added the associated citations. Thank you for pointing this out!

---

> > ### Author Response · Authors · 2022-11-12
> > **Choice of k**
> >
> > **Comment:** How do you fix k? Can it be estimated/learned? Any works that do so?
> >
> > **Response:**
> >
> > We thank the reviewer for this question. In some cases, k can be chosen using domain knowledge about the task. For example, in the wiping task, the end-effector of the robotic manipulator should always remain level with the surface of the table, only sliding on a plane. In this case it is straightforward to choose k=2. Choosing k may not be so obvious for more complex tasks. It is future work to determine how to pick an appropriate number of bases for a given task in general. The baseline, LASER, has been shown empirically in Allshire et al., 2021 to essentially discard unnecessary latent dimensions when k is chosen to be large. There is otherwise no work in this area that discusses this issue, where generally authors seek to compress the number of actions to as small a space as possible.

---

> > ### Author Response · Authors · 2022-11-12
> > **Robustness to hyperparameter choice**
> >
> > **Comment:**  Why would/should agents trained with NHT be more robust w.r.t. different hyperparameter configurations?
> >
> > **Response:**
> >
> > It is well known that exploration is a difficult open problem in reinforcement learning. With an NHT action interface an agent has the advantage of exploration in a smaller action space (i.e. lower dimensionality compared to the full DOF actuation space). In some cases, NHT eliminates the possibility of harmful or counterproductive actions. Take for example the WAMWipe environment - since NHT learns to keep the robot end effector level with the table, it is impossible for an agent to jam the arm into the table, causing excessive force and early termination of the learning episode. An agent learning in the full 7-DOF actuation space, on the other hand, could easily exhibit this unsafe behavior due to an errant exploration step.
> > In other words, an agent learning with an NHT action interface is limited to exploration only in the space of useful actuations. This is why agents with an NHT interface tend to perform well, over a wide range of hyperparameter configurations.

---

> > > ### Comment · Reviewer_2zLC · 2022-12-02
> > > **not sure**
> > >
> > > Thanks for the careful rebuttal. I am afraid I am not fully convinced by your answer. I believe the robustness to hyperparameter configurations, according to your answer, would also apply to LASER as it also reduces the dimensionality (to 2 and 3 for WAM). I would have instead expected significantly faster learning curves and other statistical improvements due to dimensionality reduction (such as better generalization etc.)

---

> > > > ### Author Response · Authors · 2022-12-10
> > > > **Robustness to hyperparameter choice 2**
> > > >
> > > > You are correct that the above argument holds for LASER as well, since it also reduces dimensionality. The analysis in our comment was incomplete; Reducing the dimensionality of the action space is not enough in itself. The learned mapping from low-dimensional actions to high-dimensional actuations must also be reasonable.
> > > >
> > > > Training NHT or LASER agents involves two steps of learning. The first step is to learn the action interface, or in other words, the mapping from low-dimensional actions to high-dimensional actuations. The second step is to train the agent that interacts with the environment using the interface.
> > > >
> > > > Our initial response to the reviewer's comment compared an NHT agent to a 7 DOF agent. Since the 7 DOF agent does not use any action interface, the answer focused on agent learning (second step) rather than the learning of the action interface (first step). In contrast, the difference between NHT and LASER lies in the first step of learning, that is, the learning of the interface itself.
> > > >
> > > > There are at least three reasons that NHT could be expected to be more robust to choice of hyperparameters: 1) NHT has fewer hyperparameters than LASER. 2) LASER has roughly twice as many learnable parameters (encoder + decoder neural networks), and thus may be more sensitive to architectural hyperparameters. 3) Learning a context-dependent linear mapping (NHT) is inherently simpler than learning a context-dependent non-linear mapping (neural network decoder of LASER). We elaborate on each of these points below.
> > > >
> > > > 1) Table 2 in the appendix lists the hyperparameters that were modulated and the ranges of values used in the experiment. We would like to point out that LASER has two regularization hyperparameters (KL loss weight and dynamics loss weight, see equation (3) in the paper). NHT has only a single regularization hyperparameter, the Lipschitz parameter. In total 7 hyperparameters were modulated for LASER, while only 6 hyperparameters were modulated for NHT. As the space of configurations grows exponentially with the number of hyperparameters, reducing the number of hyperparameters from 7 to 6 results in a significantly smaller - and therefore easier to search - hyperparameter configuration space.
> > > >
> > > > 2) In addition, LASER may have been more sensitive to architectural hyperparameters - i.e. the number of hidden layers and hidden units. These parameters affected the architecture of both the encoder and decoder neural networks in LASER. NHT has no encoder, and thus these parameters only effected a single neural network.
> > > >
> > > > 3) The complexity of the LASER model seems to empirically result in an algorithm that is sensitive to hyperparameter choice. Only a few of the hyperparameter configurations resulted in competitive performance for LASER. Our results showed that in contrast, NHT performed well with many different hyperparameter configurations. This may be explained by the fact that NHT is tasked with learning a simpler function (a context dependent linear function). In a simpler (easier) learning problem, the exact choice of hyperparameter values may be less relevant.
> > > >
> > > > We hope that this clarified why NHT was empirically more robust to hyperparameter choice compared to LASER. NHT has fewer hyperparameters, and about half as many learned parameters (one neural network instead of two) compared to LASER. Furthermore, NHT is likely more robust due to the inherent simplicity of learning linear mappings. We believe our hyperparameter search experiments were sound. Please let us know if you would like any further clarification.

---

> > ### Author Response · Authors · 2022-11-12
> > **Contribution of Section 3.1.1**
> >
> > **Comment:** Is section 3.1.1. part of the contribution?
> >
> > **Response:**
> > Exponential maps were invented and studied in the context of Riemannian geometry. In the context of learning systems, exponential maps have been previously studied in the literature on normalizing flows, and we have added these citations in a footnote in the revised submission.
> >
> > To the best of our knowledge, we are the first to use exponential maps in the context of action representation learning. As stated in our previous response, NHT itself is a novel contribution. Including the exponential map as one of the steps in NHT was a design choice to ensure NHT's Lipschitz continuity.

---

### Official Review · Reviewer_1kbF · 2022-10-25

**Confidence:** 4
**Clarity, Quality, Novelty And Reproducibility:** good
**Correctness:** 3
**Technical Novelty And Significance:** 3
**Empirical Novelty And Significance:** 3
**Recommendation:** 5

**Strength And Weaknesses:**

Strengths:
1. The proposed NHT algorithm is intriguing, with theoretical justifications, and without much compromise to make the method practical.
2. Compared to alternative methods that learn a latent action space, the proposed method appears to perform better.

Weaknesses:
1. The manipulation tasks were trained with DDPG, while newer methods like SAC or TD3 has shown notably better learning performance. It’s not clear how much impact the algorithm would still make given a stronger algorithm.
2. The comparison to using the raw action space is not entirely fair. For example in the half-cheetah case an optimal policy is already trained to collect data for learning the action representation. I think a more proper way of comparison is to separate the tasks for training the action space representation and evaluating the representation performance. For example for the two manipulation tasks one can learn the action space with one task and test on the other. For the half-cheetah task one may learn the action space with a running backward policy and test with a running forward policy.


**Summary Of The Paper:**

The paper proposed neural household transforms (NHT) for learning a low-dimensional action representation for RL problem. Prior work in this area usually learns an autoencoder-style action representation using neural networks. In this work, the authors propose to learn a network that maps to an orthogonal matrix instead of the original action space. By doing this, it enables them to avoid the need of encoding the action into the latent space as it can be obtained by an analytical least square solution. To obtain a well-behaved mapping function, exponential map is used to map the neural network output onto a unit sphere, which is then formed into the orthogonal projection matrix. The authors provided theoretical justifications for the approximation error as well as the smoothness of the proposed method. The algorithm is evaluated on two manipulation tasks and a locomotion task in simulation and is demonstrated to outperform baseline methods.

**Summary Of The Review:**

As discussed in the weaknesses, my main concerns for the work is the comparison fairness to the baseline as well as using a stronger learning algorithm for the manipulation tasks. I think the algorithm itself is novel and interesting in my knowledge and if the concerns were addressed I believe this would be a good one for the venue.

---

> ### Author Response · Authors · 2022-11-12
> **Initial Response to Reviewer 1kbF**
>
> We thank reviewer 1kbF for their insightful comments. We leave itemized responses to the concerns cited by reviewer 1kbF below. If reviewer 1kbF has any additional comments, we will provide further clarifications.

---

> > ### Author Response · Authors · 2022-11-12
> > **Choice of RL algorithm**
> >
> > **Comment:** The manipulation tasks were trained with DDPG, while newer methods like SAC or TD3 has shown notably better learning performance. It’s not clear how much impact the algorithm would still make given a stronger algorithm.
> >
> > **Response:**
> >
> > We thank the reviewer for this feedback, which concerns the learning efficiency of different reinforcement learning algorithms. We note that LASER, the latent policy framework baseline in our work, appeared to improve the performance of SAC agents in the work of Allshire et al., 2021. Across all tasks evaluated in our experiments, we saw performance gains compared to LASER.
> >
> > We also would like to point out that we used PPO for our HalfCheetah experiments, precisely to address this concern. DDPG is an off-policy algorithm, while PPO is on-policy. We have shown that NHT can improve performance in representative algorithms for both paradigms.

---

> > ### Author Response · Authors · 2022-11-12
> > **Action Representation Transfer**
> >
> > **Comment:** The comparison to using the raw action space is not entirely fair. For example in the half-cheetah case an optimal policy is already trained to collect data for learning the action representation. I think a more proper way of comparison is to separate the tasks for training the action space representation and evaluating the representation performance. For example for the two manipulation tasks one can learn the action space with one task and test on the other. For the half-cheetah task one may learn the action space with a running backward policy and test with a running forward policy.
> >
> > **Response:**
> > We agree with the reviewer that task transfer learning with action mapping methods is a promising direction for future research. NHT interfaces trained for wiping could be used with an agent learning to play air hockey (multiplayer game), learning to push blocks, or any other similarly planar task. It is an exciting research question to study action map transferability. Studying this question requires rigorous future work, which, to our knowledge, has only been limited to minor task variations in the work of Allshire et al., 2021.
> >
> > Our primary goal in this work was to focus our analysis on NHT as a new action mapping method. In our experiments we have conducted large hyperparameter sweeps to verify the efficacy of our method compared to optimized baselines.
> >
> > Furthermore, it can be realistic to assume access to a dataset of expert demonstrations without having access to the policy that generated those demonstrations. For example, in robotics kinesthetic demonstration is frequently used to collect such data. A human provides expert demonstrations, but we don’t have direct access to the policy.

---

> > > ### Comment · Reviewer_1kbF · 2022-11-21
> > > **Thanks for the response!**
> > >
> > > Thanks for the response! I agree that it is sometimes possible to obtain expert demonstration without accessing the optimal policy.
> > >
> > > My main concern is that the raw action space appears to be quite competitive in most tasks, and with expert demonstration available, it feels plausible that by doing a behavior cloning of the policy first then fine-tuning it could lead to even better performance.
> > >
> > > Thus it would be helpful to have some examples where the available data are non-optimal or not directly related to the task.

---

> > ### Author Response · Authors · 2022-11-12
> > **Summary**
> >
> > **Comment:** My main concerns for the work is the comparison fairness to the baseline as well as using a stronger learning algorithm for the manipulation tasks. I think the algorithm itself is novel and interesting in my knowledge and if the concerns were addressed I believe this would be a good one for the venue.
> >
> > **Response:**
> > We thank the reviewer for their opinion about the novelty of our algorithm. To the best of our ability, we have answered Reviewer 1KbF's concerns in our responses to the previous questions. Please let us know if Reviewer 1KbF has any additional concerns.

---

### Official Review · Reviewer_XySg · 2022-10-31

**Confidence:** 4
**Correctness:** 3
**Technical Novelty And Significance:** 2
**Empirical Novelty And Significance:** 2
**Recommendation:** 5

**Clarity, Quality, Novelty And Reproducibility:**

- The main baseline, besides the simple SVD baseline, is LASER. However, the paper does not review or explain what this baseline is. I would encourage the authors to review this method to situate their work in better context.
- Per my quick read, LASER seems to be a representative algorithm for the latent actions framework. The paper can compare against a representative algorithm from the motor program framework.
- The paper can better motivate the importance of the research question they study and specifically the advantages of a linear subspace for actions.


**Strength And Weaknesses:**

The paper proposes a nice algorithmic approach for the problem studied -- i.e. finding local linear sub-spaces for actions. However, the paper does not do a good job of motivating why this is an important problem. What are the specific advantages of linear subspaces? Why is it preferable to nonlinear per-timestep action-space mappings (nonlinear latent actions) and trajectory-level action-spaces (motor primitives)?

**Summary Of The Paper:**

This paper aims to learn a linear subspace for actions to control agents. It is well-known in the RL community that choice of action space has a dramatic effect on the learning process. For example, end-effector control in robotic manipulation is much easier than direct torque control for several tasks. Thus, learning better action spaces is an important research problem. This work specifically aims to learn local linear subspaces for actions, and utilizes NHT for efficient learning and inference. The paper also demonstrates that subspaces learned using NHT have desirable local smoothness properties.

**Summary Of The Review:**

It is unclear why this problem is important and relevant to the NeurIPS community. The authors can either motivate the problem statement better or submit to a more specialized conference where it might better appeal to the audience.

---

> ### Author Response · Authors · 2022-11-12
> **Venue**
>
> **Comment:** It is unclear why this problem is important and relevant to the NeurIPS community. The authors can either motivate the problem statement better or submit to a more specialized conference where it might better appeal to the audience.
>
> **Response:**
>
> We thank the Reviewer for their feedback, and believe we have addressed the concerns relating to the problem motivation in our previous responses. From our perspective, the International Conference on Learning Representations **(ICLR)** was an appropriate venue, since at its core, our work is about a novel method for **learning** action **representations**. We study these methods in the context of RL, one of the cornerstone areas of study in machine learning and a primary research area of interest at ICLR.

---

> ### Author Response · Authors · 2022-11-12
> **Motor Program Framework**
>
> **Comment:** Per my quick read, LASER seems to be a representative algorithm for the latent actions framework. The paper can compare against a representative algorithm from the motor program framework.
>
> **Response:**
>
> Yes, it is our intention that LASER is a representative algorithm for the latent actions framework. We have clarified this in the paper as pointed out in our previous response.
>
> If possible, could Reviewer XySg suggest literature for work in the motor program framework? We found several papers that appeared to be from the psychological literature in our investigation, but none from the machine learning community. During our literature review, the motor program framework did not come up.

---

> ### Author Response · Authors · 2022-11-12
> **LASER Baseline**
>
> **Comment:** The main baseline, besides the simple SVD baseline, is LASER. However, the paper does not review or explain what this baseline is. I would encourage the authors to review this method to situate their work in better context.
>
> **Response:**
>
> We thank the Reviewer for pointing this out. The LASER algorithm is a direct application of the Latent Policy Framework described in section 2.3, trained with Equation 3 while using all three components of the loss term. We note that it was an oversight to not mention LASER here explicitly, as we sought to position section 2.3 in the most general form to account for the variety of deep learning latent action models in the literature. We have added the following sentence to the end of section 2.3 to clarify this point:
>
> “The LASER algorithm is a representative example of this framework (Allshire et al., 2021). LASER trains a latent dynamics model in conjunction with a variational autoencoder.”

---

> ### Author Response · Authors · 2022-11-12
> **Motivation**
>
> **Comment:** The paper does not do a good job of motivating why this is an important problem. What are the specific advantages of linear subspaces? Why is it preferable to nonlinear per-timestep action-space mappings (nonlinear latent actions) and trajectory-level action-spaces (motor primitives)?
>
> **Response:**
>
> The motivation for contextual subspace approximation and the corresponding solutions can be summarized as follows:
>
> 1) Contextual subspace approximation requires less data because a *k*-dimensional subspace is completely determined by just *k* linearly independent samples.
> 2) From the agent’s perspective, action maps change the transition dynamics of the environment, and using simpler functions results in simpler dynamics.
> 3) Models for contextual subspace approximation can be notably smaller by doing away with the encoder from the latent actions framework.
>
> We have included this summary in the revised submission of our paper. We elaborate on the first two points below. We believe the third point is self-explanatory (NHT uses only one neural network while LASER or similar models in the latent policy framework use two neural networks each of comparable size).
>
> The first motivation has to do with the sample efficiency of learning the action representations themselves. If we assume that in any given context the demonstration actuations do exist on or near a *k*-dimensional subspace, then learning an actuation subspace for that context will be more sample efficient than training a neural decoder. Under this assumption, access to *k* linearly independent high-dimensional actuations uniquely determines a *k*-dimensional actuation subspace. For example, two linearly independent vectors uniquely determine a plane (2-dimensional subspace). Leveraging this inductive bias reduces the need for large datasets, which can be crucial in robotics, where demonstrations are typically expensive to collect.
>
> On the other hand, a neural decoder could satisfy the constraints implied by these *k* examples with any of an infinite number of representable functions. Which of these functions the neural decoder discovers would be some matter of chance. In order for the decoder to learn a function resembling a linear map, it would require many more samples, such that the distribution of inputs during training approximately covers the space of potential low-dimensional actions.
>
> The second motivation for learning contextual linear action representations relates to the generalization ability of an agent learning in the new low-dimensional action space. From the perspective of the agent, the mechanics involved in mapping its low-dimensional actions to the high-dimensional actuations can be considered as part of the environment dynamics. In fact, our implementation of the experiments in this paper take exactly this perspective: we wrap NHT with the base environment to create an NHT-interface version of the environment. From the agent's perspective, the action space is k-dimensional, and all of the computations of NHT are rolled into the environment dynamics. NHT is part of the environment, not part of the agent. Clearly, simpler transition dynamics in the environment can facilitate better agent generalization. Including a neural decoder action mapping (latent policy approach) in the environmental dynamics is clearly more complex than introducing a linear action mapping (NHT). Therefore, contextual linear action representations should enable better agent generalization by yielding simpler transition dynamics.
>
> Consider, for example, an agent has experience performing some action **a** in a context **c**, and the resulting high-dimensional actuation is **u**. Given a linear action map such as NHT, the high-dimensional actuation corresponding to an action 2**a** will be 2**u**, and the action -**a** will result in an actuation -**u**.
> Given the same experience, g(**a**,**c**)=**u**, with a neural decoder, could the agent guess what actuation would correspond to an action of 2**a**, or -**a**? Not unless it has direct experience with providing these exact actions in this exact context. This is the advantage of a linear action representation.

---

> ### Author Response · Authors · 2022-11-12
> **Initial Response to Reviewer**
>
> First and foremost, we thank Reviewer XySg for their insightful feedback. We leave itemized responses to Reviewer XySg's comments below. If reviewer XySg has any additional comments, we will provide further clarifications.

---

### Decision · Program_Chairs · 2023-01-20

**Decision:**

Reject

**Justification For Why Not Higher Score:**

As the e reviewers note, the claims made in the paper are not supported by adequate experimental evaluation.

**Justification For Why Not Lower Score:**

N/A

**Metareview: Summary, Strengths And Weaknesses:**

The paper considers the problem of identifying an action representation appropriate for robot manipulation tasks. The paper notes that existing methods that learn a lower-dimensional latent representation of the actions typically involve a large number of parameters. Further, the learned action representations tend to sacrifice interpretability. In an effort to overcome these limitations, the paper proposes to learn local linear subspaces for actions that are then transformed using the exponential map on the sphere to give rise to Household reflectors. The framework is supported by the underlying theoretical justification and empirically demonstrated on different locomotion and manipulation tasks for which it outperforms contemporary baselines.


While there were some questions of relevance raised in the initial reviews, the problem of learning abstract action representations is relevant and of interest to the learning community. All three reviewers appreciate the proposed algorithmic approach to identifying local linear subspaces for actions. They also appreciate the fact that the proposed method does not require fundamental compromises in order to be effective, and note the empirical evidence in support of the advantages of NHT. However, the reviewers identify a few key weaknesses with the current approach that need to be addressed in order to clarify the significance of the contributions. In particular, the reviewers question the chosen baselines as appropriate algorithms to compare against. Similarly, one reviewer points out that, despite discussions with the authors as part of the reviewing phase, they remain concerned about the conclusions regarding the hyperparameter robustness claims. Unfortunately, these concerns were not resolved as part of the discussion between the authors and two of the reviewers. The reviewers and AC agree that the paper has the potential to provide a nice contribution to the
problem of action representation learning, but the aforementioned concerns need to be addressed.

**Summary Of Ac-Reviewer Meeting:**

N/A